EMBO
*reports*

# RNF20-mediated H2B monoubiquitination protects stalled forks from degradation and promotes fork restart

Debanjali Bhattacharya [ID], Harsh Kumar Dwivedi [ID] & Ganesh Nagaraju [ID] [✉]

## Abstract

**Chromatin modifications play an important role in transcription, DNA replication and repair. Nonetheless, whether histone modifications regulate replication stress responses remains obscure. Here, we show that RNF20 localizes to and promotes H2B monoubiquitination (H2Bub) at replicating sites. Knockdown of RNF20 leads to degradation of stalled forks by nucleolytic enzymes, which can be rescued by inhibition of MRE11/DNA2 and co-depletion of SMARCAL1/HLTF/ZRANB3 fork remodelers. RNF20 facilitates the loading of RAD51 and RAD51C at stalled fork sites and acts in the same pathway of RAD51/RAD51C-mediated fork protection and restart. Analyses with RING domain and phosphorylation-deficient mutants of RNF20 show that its catalytic activity and ATR-mediated phosphorylation are essential for its role in replication stress responses. Finally, treatment of RNF20-depleted cells with chromatin relaxing agents rescues fork protection and restart defects. Collectively, our study uncovers a role for RNF20-mediated H2Bub in regulating chromatin dynamics to safeguard replicating genomes.**

**Keywords** RNF20; H2B Monoubiquitination; Replication Stress; Fork Protection; Genome Stability
**Subject Categories** Chromatin, Transcription & Genomics; DNA Replication, Recombination & Repair

## Introduction

Error-free genome duplication is critical for maintaining cellular homeostasis and disease prevention (Tubbs and Nussenzweig, 2017). DNA replication is constantly threatened by obstacles arising from both endogenous and exogenous sources. Cells have evolved with multiple mechanisms to deal with replication stress to safeguard the replicating genomes. One such mechanism is that stalled forks undergo remodeling to generate reversed forks. Such reversed forks prevent the accumulation of single-stranded DNA (ssDNA) and facilitate the repair and restart of the stalled forks (Berti et al, 2020a; Cortez, 2019; Quinet et al, 2017;

Thakar and Moldovan, 2021; Tye et al, 2021). However, the reversed forks are also susceptible to degradation by various nucleases including MRE11, EXO1 and DNA2 (Pasero and Vindigni, 2017). Failure to protect stalled forks leads to the accumulation of mutations and chromosomal aberrations, eventually leading to tumorigenesis (Aguilera and Garcia-Muse, 2013; Saxena and Zou, 2022).

Homologous recombination (HR) plays an important role in the repair of DNA double-strand breaks (DSBs), thereby preventing genome instability and suppressing tumorigenesis (Kass et al, 2016; Nagaraju and Scully, 2007; Scully et al, 2019). RAD51 recombinase nucleates onto ssDNA at the DSB sites and promotes HR. BRCA1 and BRCA2 proteins promote RAD51 filament formation, facilitating HR (Venkitaraman, 2009; Zhao et al, 2019). Mammalian genome encodes five RAD51 paralogs; RAD51B, RAD51C, RAD51D, XRCC2 and XRCC3 (Bhattacharya et al, 2022). Germline mutations in *BRCA1*, *BRCA2* and *RAD51* paralogs are known to cause breast and ovarian cancers as well as bone marrow failure syndrome Fanconi anemia (FA) (Somyajit et al, 2010, 2012; Zhao et al, 2019). RAD51 paralogs participate in the HR-mediated repair of DSBs (Nagaraju et al, 2009; Nagaraju et al, 2006; Somyajit et al, 2012), activation of intra-S-phase checkpoint (Somyajit et al, 2013) and mitochondrial genome maintenance (Mishra et al, 2018). In addition to BRCA proteins, RAD51 paralogs have been implicated as recombination mediators (Somyajit et al, 2010). Studies from various groups in the past decade demonstrated that BRCA1, BRCA2 and RAD51 paralogs have repair-independent functions in protecting stalled replication forks from nucleolytic degradation (Saxena et al, 2019; Saxena et al, 2018; Schlacher et al, 2011; Schlacher et al, 2012; Somyajit et al, 2015). In addition to these proteins, RAD52, RAD54, RIF1, CtIP, FA pathway proteins such as FANCD2, FANCJ, PALB2, human CST complex, BOD1L, ABRO1, MAD2L2 and DCAF14 proteins have been shown to protect the stalled forks (Liao et al, 2018; Rickman and Smogorzewska, 2019; Thakar and Moldovan, 2021). Nonetheless, the mechanism underlying the recruitment of these proteins to the sites of stalled forks to protect and facilitate genome duplication is unclear.

Histone posttranslational modifications (PTMs) such as ubiquitination, methylation and phosphorylation play a crucial role in gene transcription, DNA replication and repair (Chen and Tyler, 2022; Ferrand et al, 2021; Millan-Zambrano et al, 2022). The histone H2B mono-ubiquitination (H2Bub) at K120 is primarily catalyzed by RNF20/RNF40 heterodimeric E3 ubiquitin ligase in cooperation with RAD6 E2 ubiquitin

Department of Biochemistry, Indian Institute of Science, Bangalore, Karnataka 560012, India. ✉E-mail: nganesh@iisc.ac.in

conjugase (Zhu et al, 2005). H2B K120ub has been shown to regulate various cellular processes, including chromatin remodeling, transcriptional activation, RNA processing and export, DNA damage response (DDR), stem cell differentiation and tissue development (Driscoll and Yan, 2023; Kato and Komatsu, 2015; Shiloh et al, 2011). Altered/loss of expression of RNF20/RNF40 has been observed in various cancer cells, including breast, lung, prostate and renal cancers, implying the role of RNF20/40 in tumor suppression (Sethi et al, 2018).

Evidence from various studies showed that RNF20 participates in the repair of DSBs by HR, and this activity is dependent on K120 monoubiquitination of H2B by RNF20 (Moyal et al, 2011; Nakamura et al, 2011; Shiloh et al, 2011; So et al, 2019). Moreover, loss of RNF20 in the mouse germ cells affects the repair of programmed DSBs and causes male infertility (Xu et al, 2016). Knockdown of RNF20 in the germ cells affected chromatin relaxation and recruitment of repair factors to the sites of DSBs (Xu et al, 2016). Similarly, *Saccharomyces cerevisiae* cells lacking Bre1, an ortholog of human RNF20, showed defective repair of DSBs by HR and impaired loss of histones spanning the DSB sites (Zheng et al, 2018). A recent study showed that Bre1-mediated H2Bub is essential for D-loop formation and its stabilization via regulating chromatin dynamics (Hung et al, 2025). These studies suggest an important role of RNF20-mediated H2Bub in chromatin decompaction in facilitating HR-mediated repair. In addition to DSB repair, RNF20/Bre1 participates in DNA replication and DNA damage tolerance (DDT) pathways to facilitate error-free genome duplication (Hung et al, 2017; Liu et al, 2021; Trujillo and Osley, 2012). Bre1 localizes to chromatin, spanning the origins, and catalyzes H2Bub to promote DNA replication and nucleosome assembly (Trujillo and Osley, 2012). Localization of Bre1 to the replication-coupled damage sites promotes lesion bypass and fork recovery by HR (Hung et al, 2017; Liu et al, 2021; Northam and Trujillo, 2016). Cells lacking Bre1/RNF20 accumulate replication-associated DNA lesions and display increased micronucleation, chromosome instability and cell death (Chernikova et al, 2012; Hung et al, 2017; Liu et al, 2021). However, the underlying mechanism by which RNF20 participates in replication stress responses and genome maintenance is less understood.

Here, we find that RNF20-deficient cells exhibit increased replication stress markers such as micronuclei and 53BP1 nuclear bodies. RNF20 accumulates at the sites of stalled replication forks and promotes H2Bub at K120. Depletion of RNF20 leads to the degradation of stalled forks by MRE11/DNA2 nuclease, which can be rescued by inhibition of MRE11/DNA2 and co-depletion of fork remodelers. RNF20 knockdown affects RAD51 and RAD51 paralogs recruitment to stalled replication sites, resulting in fork degradation and impaired fork recovery. Mechanistic studies reveal that ATR-mediated phosphorylation of RNF20, and its catalytic activity are essential for replication stress responses. Strikingly, treatment of RNF20-deficient cells with chromatin relaxing agents rescues the fork protection and restart defects. Together, our studies demonstrate that RNF20-mediated H2Bub promotes chromatin decompaction at stalled fork sites and facilitates error-free genome duplication.

# Results

## RNF20 is required for preserving genomic integrity during replication stress

Studies from yeast and mouse model systems indicate that RNF20 contributes to DNA replication, repair of replication-associated

DNA damage and fork recovery (Chernikova et al, 2012; Lin et al, 2014; Northam and Trujillo, 2016). However, the precise function of RNF20 during replication stress in mammalian cells remains obscure. To investigate this, we generated two shRNAs; one corresponding to UTR (shRNF20 #1) and one gene-specific (shRNF20 #2) and examined replication stress markers in the control and RNF20-depleted U2OS human osteosarcoma cells. A standard marker of replication stress is common fragile site (CFS) instability, a hallmark found in many cancer cells (Durkin and Glover, 2007). CFS expression in the form of breaks and gaps occurs due to intrinsic replication stress. Such lesions are marked by 53BP1 nuclear bodies in the G1 phase cells, which are passed on to daughter cells from the previous cell cycle (Harrigan et al, 2011). Interestingly, the knockdown of RNF20 led to an increase in 53BP1 nuclear bodies in U2OS cells with aphidicolin (APH) treatment, a potent inducer of CFS instability (Fig. 1A–C) (Durkin and Glover, 2007). Another marker of CFS instability under replication stress is the appearance of micronuclei, small extra-nuclear bodies containing chromosomal fragments that are mis-segregated during cell division. We observed that RNF20-depleted cells accumulated a higher percentage of micronucleated cells than control cells, and this was further increased upon APH treatment (Fig. 1D,E). Consistently, FANCD2 foci in mitotic cells, a marker for under-replicated genomic regions (Chan et al, 2009), were elevated in RNF20-depleted cells under APH treatment (Fig. EV1A,B). Replication stress causes uncoupling of helicase and polymerases, exposing extensive single-stranded DNA (ssDNA) stretches which are coated by the heterotrimeric ssDNA binding protein, Replication Protein A (RPA). RNF20-depleted cells accumulated higher levels of RPA70/pRPA32 S4/8 than control cells in response to hydroxyurea (HU) treatment (Figs. 1F,G and EV1C,D). In addition, ssDNA accumulation, scored by BrdU foci, was also elevated in RNF20-depleted cells compared to the control cells (Fig. EV1E,F). Prolonged replication stress leads to the collapsing of stalled replication forks into DSBs. RNF20 knockdown led to higher γH2AX foci formation, a marker for DSBs, and an increased comet tail moment when treated with HU (Figs. 1H,I and EV1G,H). To further assess the role of RNF20 in preserving genomic integrity, we investigated gross chromosomal abnormalities in the form of breaks, radials and gaps generated in RNF20-depleted cells under replication stress. In agreement with our previous observations, the loss of RNF20 led to increased chromosomal aberrations (Fig. EV1I,J). RNF20-depleted cells also exhibited considerable sensitivity to HU- and APH-induced replication stress (Fig. 1J,K). These results establish the importance of RNF20 in preventing replication stress-associated catastrophes and maintaining genomic stability during replication stress.

## RNF20 localizes to and mediates histone H2B monoubiquitination at stalled fork sites

A large-scale iPOND proteomics screen previously reported that RNF20 is enriched on nascent DNA at replication forks (Wessel et al, 2019). To investigate the localization of RNF20 at replication fork sites, we performed iPOND (isolation of Proteins on Nascent DNA) in actively replicating, stalled and thymidine chase (mature chromatin) conditions (Fig. 2A) and investigated the status of different proteins by western blotting. RNF20 was found to be associated with active forks, and it was enriched at stalled fork sites. Additionally, RAD51 and RAD51C were also

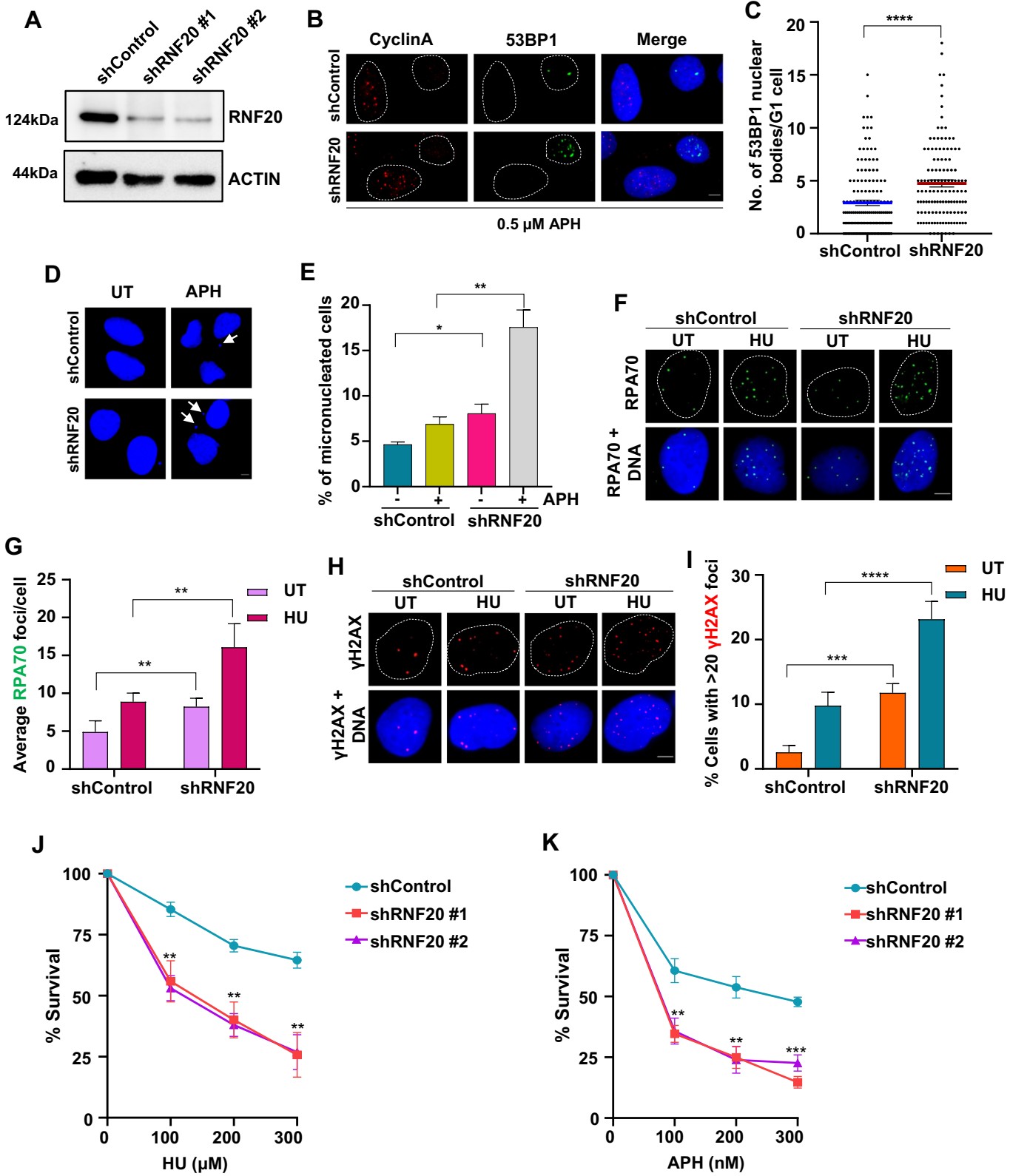

**Figure 1. RNF20 is essential for maintaining genomic integrity during replication stress.**

(A) Representative immunoblot showing knockdown of RNF20 in U2OS cells using two independent shRNAs. #1 indicates a UTR specific and #2 indicates a gene-specific shRNA against RNF20. Actin serves as a loading control. (B) Representative images showing 53BP1 nuclear bodies in cyclin A negative (G1 phase) U2OS cells treated with 0.5 μM APH for 12 h along with the indicated shRNAs. Scale bar = 5 μm. Dotted white line indicates the nuclear boundary of cells. (C) Scatter plot showing the number of 53BP1 nuclear bodies per cyclin A negative U2OS cell as shown in (B). Data represent mean ± SEM from three independent experiments. Total of ≥150 cells were analyzed for each condition. Mann–Whitney t test, ****$P < 0.0001$. $P$ (shRNF20 vs. shControl) < 0.0001. (D) Representative images of micronuclei formation in control and RNF20-depleted U2OS cells with or without 0.5 μM APH treatment for 12 h. Scale bar = 5 μm. White arrows indicate micronuclei. (E) Bar graph showing the percentage of micronucleated cells as shown in (D). Data represent mean ± SEM from three independent experiments. Total of ≥300 cells were analyzed for each condition. Unpaired t test, *$P < 0.05$, **$P < 0.01$. $P$ (shRNF20 HU vs. shControl HU) = 0.0067, $P$ (shRNF20 UT vs. shControl UT) = 0.033. (F) Representative images of RPA70 foci formation in control and RNF20-depleted U2OS cells in untreated (UT) condition or following a recovery of 6 h after HU (4 mM, 2 h) treatment. Scale bar = 5 μm. Dotted white line indicates the nuclear boundary of cells. (G) Quantification of average RPA70 foci per cell in cells as indicated in (F). Data represent mean ± SD from three independent experiments. Total of ≥300 cells were analyzed for each condition. Two-way ANOVA, **$P < 0.01$. $P$ (shControl HU vs. shRNF20 HU) = 0.0098, $P$ (shControl UT vs. shRNF20 UT) = 0.0098. (H) Representative images showing γH2AX foci formation in control and RNF20-depleted cells in UT condition or following 6 h recovery after HU (4 mM, 2 h) treatment. Scale bar = 5 μm. Dotted white line indicates the nuclear boundary of cells. (I) Bar graph showing the percentage of cells with greater than 20 γH2AX foci in cells as indicated in (H). Data represent mean ± SD from three independent experiments. Total of ≥300 cells were analyzed for each condition. Two-way ANOVA, ***$P < 0.001$, ****$P < 0.0001$. $P$ (shControl UT vs. shRNF20 UT) = 0.0008, $P$ (shControl HU vs. shRNF20 HU) < 0.0001. (J) Plot showing survival of HeLa cells transfected with the indicated shRNAs and subjected to continuous treatment of HU for 5 days. Cell survival was determined by MTT assay. Data represents mean ± SD. Unpaired t test, **$P < 0.01$, ***$P < 0.001$. $P$ (shControl vs. shRNF20 #1) = 0.0047, 0.0025, 0.0023 for 100 μM, 200 μM, 300 μM HU, respectively, $P$ (shControl vs. shRNF20 #2) = 0.0007, 0.0005, 0.0012 for 100 μM, 200 μM, 300 μM HU respectively. (K) Plot showing survival of HeLa cells transfected with the indicated shRNAs and subjected to treatment of APH for 48 h. Cell survival was determined by MTT assay. Data represents mean ± SD. Unpaired t test, **$P < 0.01$, ***$P < 0.001$. $P$ (shControl vs. shRNF20 #1) = 0.0017, 0.0014, 0.0001 for 100 nM, 200 nM, 300 nM APH, respectively; $P$ (shControl vs. shRNF20 #2) = 0.0042, 0.0018, 0.0003 for 100 nM, 200 nM, 300 nM APH, respectively. Wherever not explicitly stated, shRNA #1 has been used for RNF20 knockdown. Source data are available online for this figure.

detected prominently at stalled forks (Fig. 2B). RPA70, PCNA, H3 and H2B were used as controls (Dungrawala et al, 2015; Genois et al, 2021; Sirbu et al, 2012). These observations indicate that RNF20 is associated with active forks and is enriched upon stalling of replication forks.

We corroborated our finding in parallel with in-situ analysis of protein interactions at DNA replication forks (SIRF), a recently developed proximity ligation assay (PLA) system for quantitative, sensitive, and effective detection of protein association with the replication fork (Roy et al, 2018). Asynchronous U2OS cells were pulse labelled with a thymidine analog 5′-ethynyl-2′-deoxyuridine (EdU) for 8 min (active forks) followed by either HU treatment (fork stalling) or thymidine chase (mature chromatin) for 4 h. Using click chemistry, a biotin moiety was conjugated to the EdU molecules. This allowed us to visualize the localization of RNF20 at the replication sites under different conditions using specific antibodies against biotin and RNF20. In agreement with our iPOND analysis, RNF20 SIRF signals were found at the actively progressing forks and it was significantly elevated at the stalled fork sites (Fig. 2C,D).

RNF20 monoubiquitinates histone H2B at lysine 120 (H2B K120ub) to facilitate transcription and the recruitment of DNA damage response (DDR) proteins (Moyal et al, 2011; Nakamura et al, 2011; Pavri et al, 2006). We investigated whether RNF20 also ubiquitinates H2B at actively progressing and stalled replication fork sites by performing SIRF in control and RNF20-depleted cells using H2B K120ub specific antibody. H2B K120ub SIRF signals were detected at progressing replication forks, and these signals were further enriched at stalled replication forks in the control cells (Figs. 2E,F and EV2A). However, the H2B K120ub SIRF signals were abrogated at both progressing and stalled replication forks upon depletion of RNF20 (Fig. 2E,F), indicating that RNF20 catalyzes H2B monoubiquitination at fork sites.

## RNF20 is required for stability of stalled forks and their recovery

The accumulation of RNF20 at stalled fork sites prompted us to investigate the function of RNF20 at stalled replication forks. We

employed the DNA fiber assay to analyze fork protection and restart dynamics in control and RNF20-depleted cells under replication stress conditions. U2OS cells were sequentially pulsed with 5′-chloro-2′-deoxyuridine (CldU) and 5′-iodo-2′-deoxyuridine (IdU) and subjected to prolonged replication stress by HU treatment. Reduced IdU (green) to CldU (red) tract length ratios revealed a severe fork protection defect in RNF20-depleted cells (Fig. 3A,B). This fork protection defect was significantly rescued by treating cells with mirin (Figs. 3C,D and EV2B), an inhibitor of MRE11 nuclease, which degrades nascent DNA at reversed forks. In addition, fork stabilization defect in RNF20-depleted cells was also rescued by inhibiting DNA2 nuclease with C5 inhibitor (Fig. 3E,F). Fork reversal occurs through the concerted action of RAD51 and SNF2-family fork remodelers like SMARCAL1, ZRANB3, HLTF as well as FBH1 (Bhattacharya et al, 2022; Liu et al, 2020; Liu et al, 2023; Taglialatela et al, 2017; Tye et al, 2021; Zellweger et al, 2015). Co-depletion of SMARCAL1/ZRANB3/ HLTF rescued the fork protection defect in the RNF20-depleted cells (Figs. 3G,H and EV2C–E). Notably, co-depletion of RNF20 with BRCA2, a major factor in the FA/BRCA pathway of fork protection (Schlacher et al, 2011), did not exacerbate the fork degradation compared to BRCA2 alone depleted cells (Fig. EV2F,G). Furthermore, RNF20-depleted cells showed a significant defect in fork restart following recovery from replication stress with a concomitant increase in stalled forks compared to the control cells (Fig. 3I). However, the percentage of new origin firing events remained unaltered (Fig. 3I). Together, these results clearly indicate the role of RNF20 in protecting the stalled forks from nucleolytic degradation and facilitating fork restart during replication stress.

## RNF20 and the RAD51 paralogs participate in a common pathway of stalled fork protection and restart

Previous studies have demonstrated the roles of RAD51 paralogs (RAD51B, RAD51C, RAD51D, XRCC2 and XRCC3) in the protection and restart of stalled forks during replication stress

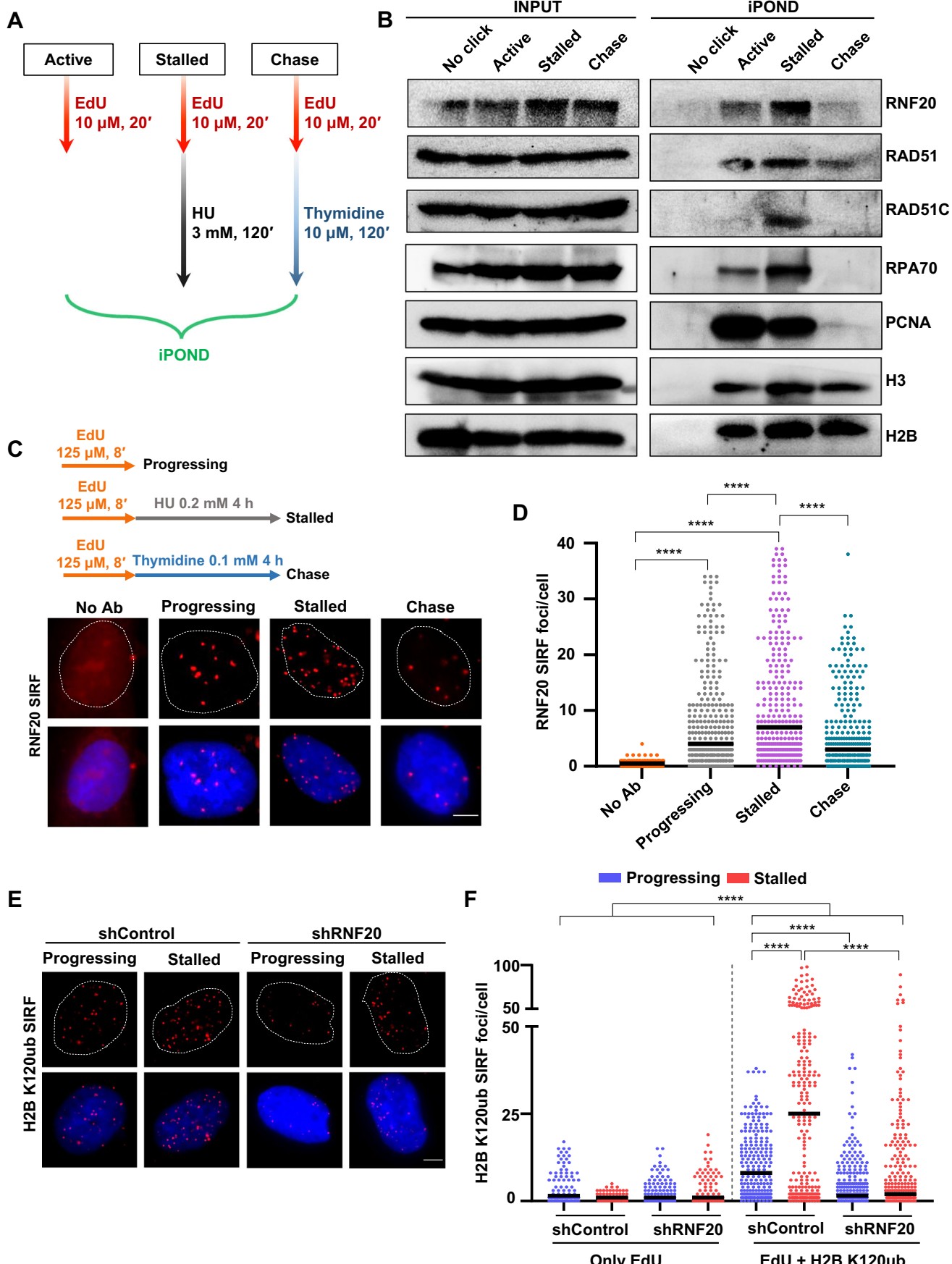

**Figure 2. RNF20 is recruited to stalled fork sites and promotes histone H2B monoubiquitination.**

(A) Schematic for EdU-iPOND experiment. Exponentially growing HeLa cells were pulse-labelled with 10 µM EdU for 20 mins (active) followed by either 3 mM HU for 2 h (stalled) or 10 µM thymidine for 2 h (chase) and iPOND was performed as described in methods. (B) Western blots showing inputs and enrichment of indicated proteins at active and stalled replication forks and post-replicative DNA (mature chromatin) from iPOND assay. (C) Representative images of RNF20 SIRF signals in asynchronous U2OS cells treated with EdU ± HU/thymidine for 4 h. No Ab: no antibody is a negative control. Scale bar = 5 µm. Dotted white line indicates the nuclear boundary of cells. (D) Quantitative scatter plot of RNF20 SIRF signals in conditions as shown in (C). Total of ≥300 cells were analyzed for progressing, stalled and thymidine chase conditions from three independent experiments. ≥150 cells were counted for no Ab. Black bars represent median values. Mann–Whitney $t$ test, ****$P < 0.0001$. Exact $P$ value for all the indicated comparisons is $P < 0.0001$. (E) Representative images of H2B K120ub SIRF signals at progressing and stalled forks in U2OS cells transfected with shControl or shRNF20. Scale bar = 5 µm. Dotted white line indicates the nuclear boundary of cells. (F) Quantitative scatter plot of H2B K120ub SIRF signals in conditions as shown in (E). Total of ≥225 cells were analyzed for each condition from three biological repeats. Total of ≥100 cells were counted for the negative control samples. Black bars represent median values. Mann–Whitney $t$ test, ****$P < 0.0001$. Exact $P$ value for all the indicated comparisons is $P < 0.0001$. Source data are available online for this figure.

(Guh et al, 2023; Saxena et al, 2019; Saxena et al, 2018; Somyajit et al, 2015). RAD51 paralogs exist in two major complexes – BCDX2 and CX3, and these complexes synergize with RAD51 in protecting stalled replication forks against the action of nucleolytic enzymes (Guh et al, 2023). However, the CX3 complex, but not the BCDX2 complex, exclusively participates in the fork restart (Berti et al, 2020b; Somyajit et al, 2015). Since our results showed RNF20's role in fork protection and restart, we were curious to investigate whether RNF20 and the RAD51 paralogs perform their fork protection functions in an epistatic manner. To test this, we examined the nascent strand degradation in RNF20 and RAD51 paralogs co-depleted cells and compared it with RNF20/RAD51 paralogs alone depleted cells. Notably, while single depletions of RNF20, RAD51C, XRCC2 and XRCC3 each resulted in a significant defect in fork protection, co-depletion of RNF20 with either one of the RAD51 paralogs did not further exacerbate this effect (Figs. 4A and EV2H,I). We next examined the status of fork restart in RNF20 and RAD51C/XRCC3 co-depleted cells by performing the fork restart assay. Interestingly, the knockdown of RAD51C or XRCC3 in the background of RNF20 depletion did not alter the percentage of stalled and restarted forks when compared with RNF20/RAD51C/XRCC3 single-depletion samples (Figs. 4B and EV2J,K). Together, these observations establish that RNF20 participates in the same pathway of RAD51 paralogs in the protection and restart of stalled replication forks during the replication stress response.

The epistatic relationship between RNF20 and the RAD51 paralogs in the replication stress responses made us ask which factor is upstream in this pathway. To decipher this, we analyzed the localization of RNF20 at progressing and stalled replication forks in RAD51C/RAD51-depleted cells by SIRF assay. Interestingly, RNF20 SIRF signals remained unchanged in RAD51C/RAD51-depleted cells compared to control cells (Figs. 4C,D and EV2L,M), indicating that RNF20 localization to replication fork sites is independent of RAD51C/RAD51. We verified these results with an immunofluorescence (IF) technique, where we measured the global recruitment of flag-tagged, wild-type (WT) RNF20 (expressed ectopically) to chromatin in RAD51C-deficient cells. In accordance with our SIRF data, the average number of flag-RNF20 foci per cell remained unchanged between control and RAD51C-deficient cells in unperturbed or HU-treated conditions (Fig. EV3A,B).

Next, we analyzed the recruitment of RAD51C/RAD51 to replication fork sites in the absence of RNF20. Notably, the number of RAD51C/RAD51 SIRF foci at stalled replication forks was significantly reduced in RNF20-depleted cells compared to the control cells (Figs. 4E,F and EV3C,D). In parallel, we examined the global recruitment of RAD51C/RAD51 to chromatin in RNF20-deficient cells by IF experiments. Our results revealed a stark reduction in RAD51C/RAD51 foci per cell in HU-treated, RNF20-depleted cells compared to the control cells (Fig. EV3E–H). In addition, the localization of BRCA2 to stalled fork sites was also impaired upon RNF20 depletion (Fig. EV3I,J). Collectively, these data suggest that RNF20 functions upstream to BRCA2/RAD51/RAD51 paralogs during replication stress and facilitates their recruitment to stalled fork sites to promote the FA-BRCA pathway of fork protection and subsequent restart.

## E3 ubiquitin ligase activity of RNF20 is essential for fork protection and restart

RNF20-mediated H2B K120ub has diverse functions, including transcriptional activation and DSB repair (Moyal et al, 2011; Nakamura et al, 2011; Pavri et al, 2006). RNF20 possesses multiple coiled-coil domains and a RING domain at the C-terminus, which is essential for its E3-ubiquitin ligase activity (Fig. 5A) (Foglizzo et al, 2016). To investigate the catalytic functions of RNF20 in replication stress responses, we mutated two critical cysteine residues in the RNF20 RING domain, C922S and C960A, by site-directed mutagenesis (Fig. 5A). We depleted endogenous RNF20 in U2OS cells using a UTR-specific shRNA and simultaneously expressed the flag-tagged wild-type (WT) and RNF20 mutants. The abundance of RNF20 mutants was comparable to WT RNF20 (Fig. 5B). On examining the fork protection with the DNA fiber assay, we observed that the fork degradation in RNF20-deficient cells could be rescued with the expression of the WT RNF20 but not with the C922S and C960A mutants (Fig. 5C,D). A similar result was obtained in the fork restart assay, where the percentage of stalled and restarted replication forks were rescued by expression of WT RNF20 in RNF20-depleted cells but not with the catalytically inactive mutants of RNF20 (Fig. 5E,F). These results indicate the requirement of E3-ubiquitin ligase activity of RNF20 in the stabilization and restart of stalled replication forks. In agreement with this, H2B K120ub at the stalled forks was abrogated in cells expressing RNF20 RING domain mutants compared to WT RNF20 expressing cells (Fig. EV4A,B). However, the localization of RNF20 RING domain mutants to fork sites was unaffected compared to WT RNF20 (Fig. EV4C,D). These observations further indicate that fork protection and restart defects in cells expressing catalytically inactive RNF20 are attributed to the E3-ligase activity of RNF20.

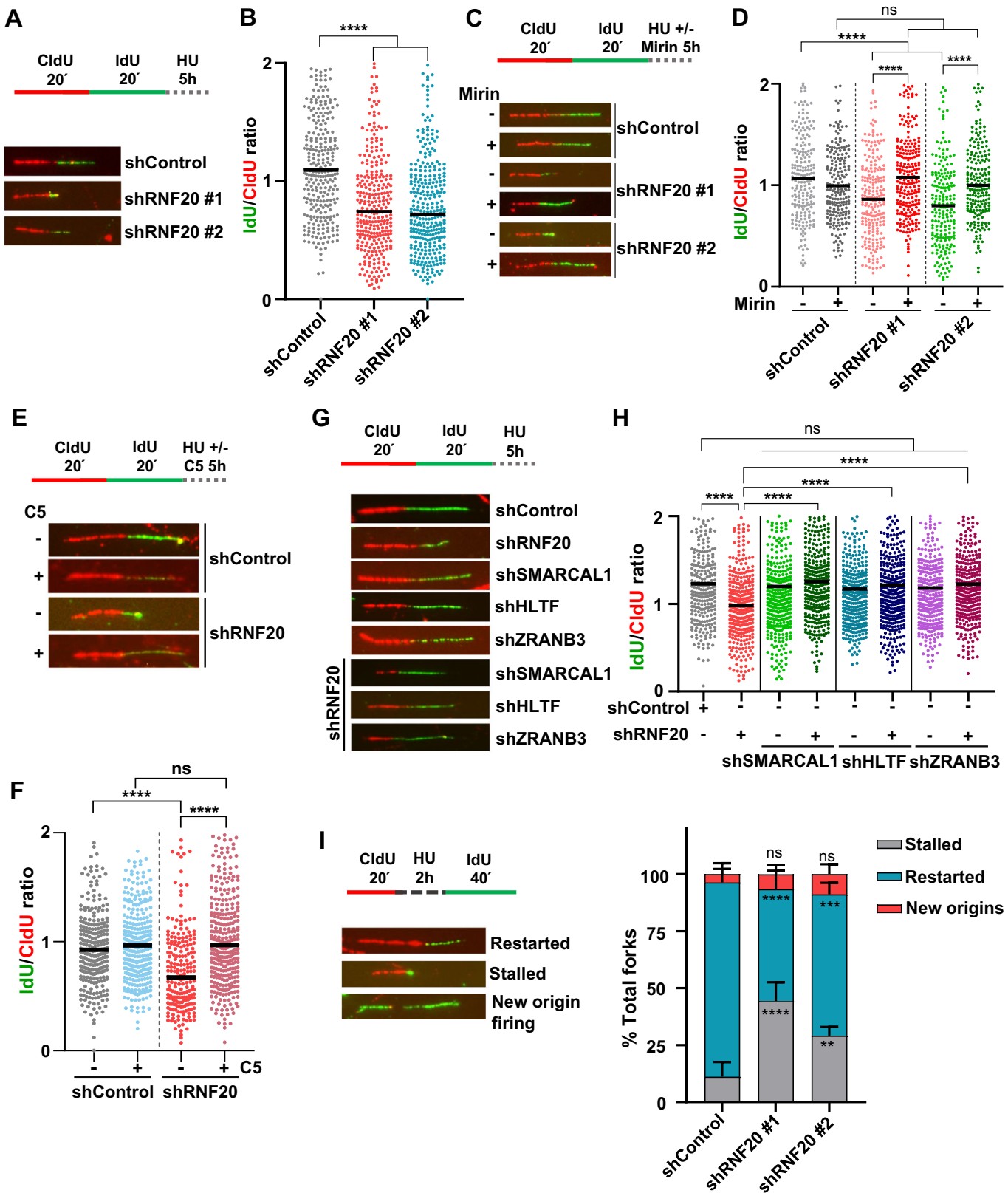

**Figure 3. Loss of RNF20 impairs stalled fork stability and fork recovery.**

(A) Representative DNA fibers showing fork degradation in control and RNF20-depleted U2OS cells treated with 4 mM HU for 5 h. (B) Scatter plot showing quantification of IdU to CldU tract length ratio in cells as shown in (A). Total of ≥300 fibers were analyzed for each condition from three independent experimental repeats. Black bars represent median values. Mann–Whitney test, ****$P < 0.0001$. p(shControl vs. shRNF20 #1; shControl vs. shRNF20 #2) < 0.0001. (C) Representative DNA fibers showing fork degradation in the indicated U2OS cells treated with HU ± Mirin for 5 h. (D) Quantification of IdU to CldU tract length ratio in cells as shown in (C). Total of ≥200 fibers were analyzed for each condition from three independent experimental repeats. Black bars represent median values. Mann–Whitney test, ****$P < 0.0001$; ns, non-significant. $P$ (shControl vs. shRNF20 #1; shControl vs. shRNF20 #2) < 0.0001, $P$ (shRNF20 #1 vs. shRNF20#1 + Mirin) < 0.0001, $P$ (shRNF20 #2 vs. shRNF20#2 + Mirin) < 0.0001, $P$ (shControl + Mirin vs. shRNF20 #1 + Mirin) = 0.2863, $P$ (shControl + Mirin vs. shRNF20 #2 + Mirin) = 0.8646. (E) Representative DNA fibers showing fork degradation in the indicated U2OS cells treated with 4 mM HU for 5 h with or without treatment with DNA2 inhibitor C5. (F) Quantification of IdU to CldU tract length ratio in cells as shown in (E). Total of ≥200 fibers were analyzed for each condition from three experimental repeats. Black bars represent median values. Mann–Whitney test, ****$P < 0.0001$; ns, non-significant. $P$ (shControl vs. shRNF20; shRNF20 vs. shRNF20 + C5) < 0.0001, $P$ (shControl + C5 vs. shRNF20 + C5) = 0.4262. (G) Representative DNA fibers showing fork degradation in the indicated U2OS cells treated with 4 mM HU for 5 h. (H) Scatter plot showing quantification of IdU to CldU tract length ratio in cells as shown in (G). Total of ≥250 fibers were analyzed for each condition from three independent experimental repeats. Black bars represent median values. Mann–Whitney test, ****$P < 0.0001$; ns, non-significant. $P$ (shControl vs. shRNF20; shRNF20 vs. shRNF20 + shSMARCAL1; shRNF20 vs. shRNF20 + shZRANB3; shRNF20 vs. shRNF20 + shHLTF) < 0.0001, $P$ (shControl vs. shRNF20 + shSMARCAL1; shControl vs. shRNF20 + shZRANB3; shControl vs. shRNF20 + shHLTF) > 0.9999. (I) Quantification of stalled, restarted forks and new origin firing events in control and RNF20-depleted cells after release from 2 mM, 2 h HU treatment. Examples of various types of tracts are shown in the left panel. Stalled and restarted replication forks are shown as a percentage of all CldU-labeled tracks. Total of ≥250 fibers were analyzed for each condition from three independent experiments. Data is represented as mean + SD. Two-way ANOVA, **$P < 0.01$; ***$P < 0.001$; ****$P < 0.0001$; ns, non-significant. $P$ (shControl stalled vs. shRNF20 #1 stalled) <0.0001, $P$ (shControl restarted vs. shRNF20 #1 restarted) <0.0001, $P$ (shControl stalled vs. shRNF20 #2 stalled) = 0.0048, $P$ (shControl restarted vs. shRNF20 #2 restarted) = 0.0005. DNA fiber labeling protocol has been indicated for each panel. Source data are available online for this figure.

RNF20-mediated H2Bub has been reported to disrupt higher-order chromatin structure and cause chromatin relaxation to facilitate transcription elongation and the recruitment of different DNA repair factors during DSB repair (Fierz et al, 2011; Nakamura et al, 2011; Oliveira et al, 2014). During replication stress, the regressed ends of reversed forks have been shown to undergo regular chromatinization (Schmid et al, 2018). Our data showed that RNF20-mediated H2Bub is required for RAD51 and RAD51C localization to stalled replication forks for its protection and restart. To gain insights into the mechanism of RNF20-mediated replication stress responses, we analysed the levels of different chromatin decompaction markers like H3K27 acetylation (H3K27ac) and H3K4 tri-methylation (H3K4me3) at the stalled fork sites by the SIRF assay. Both these chromatin marks were found to be enriched at stalled forks in control cells. Interestingly, there was a drastic reduction in H3K27ac level (Fig. 5G,H) and a modest but significant decrease in H3K4me3 level (Fig. EV4E,F) upon RNF20 depletion. Further, we examined whether the impaired fork protection in RNF20-knockdown cells can be rescued by nucleo-some relaxation. To test this, we treated cells with chloroquine and trichostatin A (TSA), which are reported to relax the chromatin, and studied fork protection (Nakamura et al, 2011; Oliveira et al, 2014). Interestingly, treatment of RNF20 knockdown cells with chloroquine or TSA rescued the fork protection defect (Figs. 5I and EV4G,H), indicating that chromatin relaxation bypasses the requirement of RNF20 in fork protection. Consistently, chloroquine or TSA also rescued fork restart defects in RNF20-deficient cells (Figs. 5J and EV4G,I).

Since RAD51/RAD51C loading at sites of replication stalling was defective in RNF20-deficient cells, we investigated whether chromatin relaxation can rescue this defect. To test this, we performed both SIRF and iPOND studies to analyse the localization of RAD51 and RAD51C to stalled forks in RNF20-deficient cells upon treatment with chloroquine or TSA. Indeed, treatment with either chloroquine or TSA rescued the recruitment of RAD51 (Fig. EV5A,B) and RAD51C (Fig. EV5C,D) to the stalled fork sites in the RNF20-depleted cells. Consistently, our iPOND data showed

a decrease in RAD51/RAD51C levels in RNF20-deficient cells, which was rescued with chloroquine treatment (Fig. EV5E). These results suggest that RNF20-mediated H2Bub is required for chromatin relaxation at stalled fork sites to recruit RAD51/RAD51C to promote fork protection and restart.

## RNF20 phosphorylation by ATR is essential for efficient fork protection and restart

ATR and ATM master kinases regulate replication stress responses and DNA damage signaling by phosphorylating numerous proteins at SQ/TQ motifs (Matsuoka et al, 2007). RNF20 has been shown to undergo phosphorylation at S172 and S553 residues by ATM in response to DSBs, and this phosphorylation is required for H2Bub (Moyal et al, 2011). We wanted to investigate whether RNF20 undergoes phosphorylation in response to replication stress by ATR. To test this, we immunoprecipitated endogenous RNF20 from mock or HU-treated U2OS cells and measured its phosphorylation by western blotting. Interestingly, HU-induced replication stress led to a stark increase in RNF20 phosphorylation, as detected with an anti-(pS/pT)Q antibody (Fig. 6A). Notably, RNF20 also interacted with RPA70 in both mock and HU-treated conditions. The lack of interaction between RNF20 and β-ACTIN served as a negative control (Fig. 6A). Strikingly, treatment with an ATR but not an ATM inhibitor abolished this phosphorylation (Fig. 6B), indicating that RNF20 undergoes phosphorylation in an ATR-dependent manner in response to replication stress. Next, we generated potential ATR target site mutations in RNF20 (S172A and S553A) (Fig. 6C). The cellular levels of RNF20 phosphomutants were comparable to that of WT RNF20 (Fig. 6C). To examine the role of RNF20 phosphorylation, we analyzed the efficiency of RNF20 phosphomutants in protecting the stalled replication forks. Reduced IdU to CldU ratios indicated inefficient fork protection in cells expressing S172A and S553A RNF20 compared to control and WT RNF20 expressing cells (Fig. 6D,E). We also examined the percentages of stalled and restarted forks in these cells by the fork restart assay and found fork restart defects in RNF20 phosphomutant-expressing cells compared to WT cells (Figs. 6F and EV5F). We further

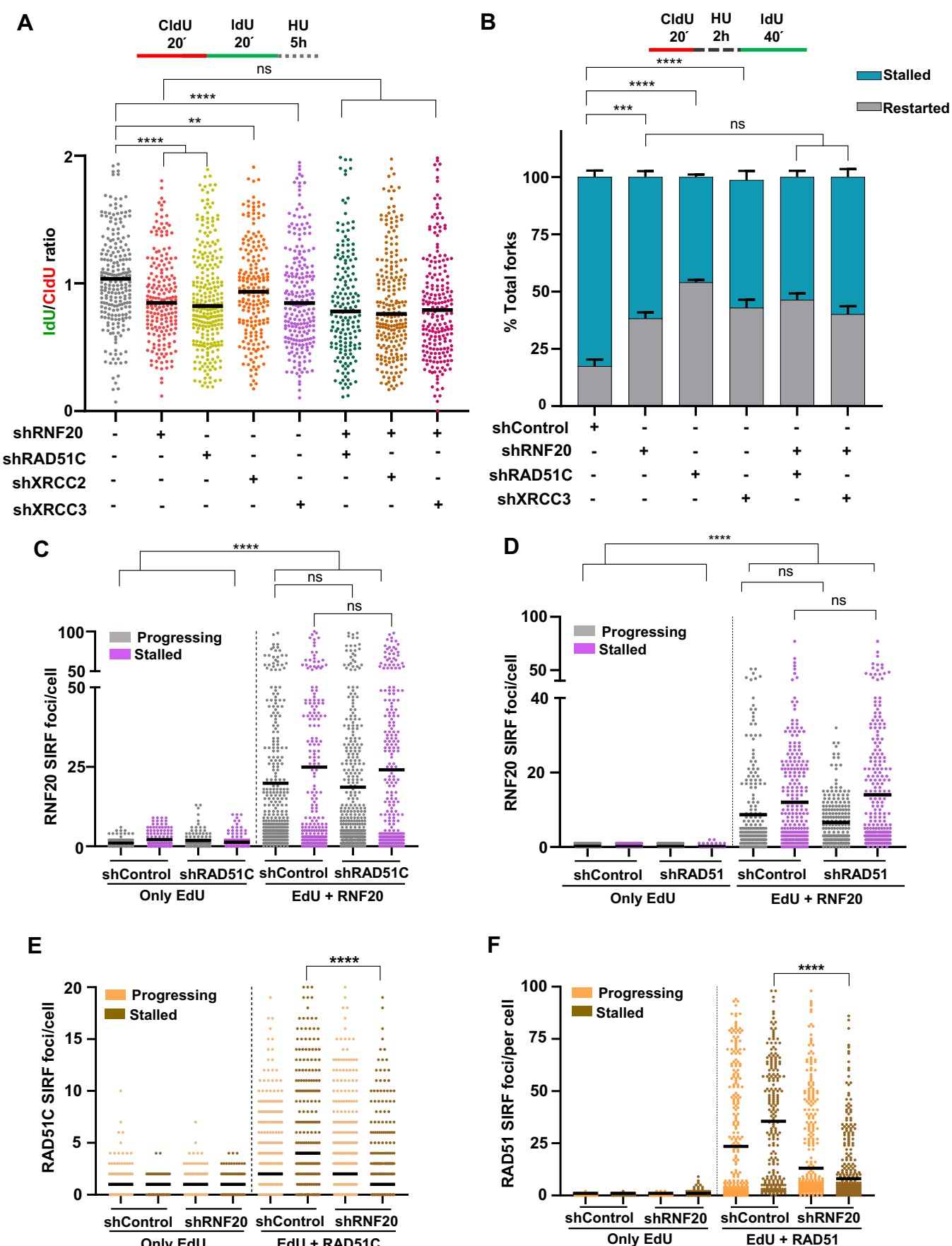

**Figure 4. RNF20 and RAD51 paralogs function in a common pathway of stalled fork protection and restart.**

(A) DNA fiber labeling protocol (top). Quantification of IdU to CldU tract length ratios in RNF20 and RAD51 paralogs depleted cells treated with 4 mM HU for 5 h. Total of ≥200 fibers were analyzed for each condition from three independent experiments. Black bars represent median values. Mann–Whitney $t$ test, **$P < 0.01$; ****$P < 0.0001$; ns, non-significant. $P$ (shControl vs. shRNF20; shControl vs. shRAD51C; shControl vs. shXRCC3) < 0.0001, $P$ (shControl vs. shXRCC2) = 0.0026, $P$ (shRNF20 vs. shRNF20 + shRAD51C) = 0.1048, $P$ (shRNF20 vs. shRNF20 + shXRCC2) = 0.1622, $P$ (shRNF20 vs. shRNF20 + shXRCC3) = 0.0955. (B) DNA fiber labeling protocol (top). Quantification of the percentage of stalled and restarted forks in RNF20 and RAD51 paralogs depleted cells. Stalled and restarted replication forks are shown as a percentage of all CldU-labeled tracks. Total of ≥250 fibers were analyzed for each condition. Data is represented as mean + SEM from three independent experiments. Two-way ANOVA, ***$P < 0.001$; ****$P < 0.0001$; ns, non-significant. $P$ (shControl stalled vs. shRNF20 stalled) = 0.0004, $P$ (shControl restarted vs. shRNF20 restarted) = 0.0004, $P$ (shControl vs shRAD51C; shControl vs. shXRCC3) < 0.0001, $P$ (shRNF20 vs. shRNF20 + shRAD51C) = 0.3977, $P$ (shRNF20 vs. shRNF20 + shXRCC3) = 0.9977. (C) Scatter plot of RNF20 SIRF signals at progressing and stalled replication forks in control and RAD51C-depleted cells. Total of ≥250 cells were analyzed for each condition. Black bars represent mean from three biological repeats. Mann–Whitney $t$ test, ****$P < 0.0001$; ns, non-significant. $P$ (shControl progressing vs. shRAD51C progressing) = 0.1468, $P$ (shControl stalled vs. shRAD51C stalled) = 0.5005. (D) Quantitative scatter plot of RNF20 SIRF signals at progressing and stalled replication forks in control and RAD51-depleted cells. Total of ≥200 cells were analyzed for each condition. Black bars represent mean from three independent experiments. Mann–Whitney $t$ test, ****$P < 0.0001$; ns, non-significant. $P$ (shControl progressing vs. shRAD51 progressing) = 0.7283, $P$ (shControl stalled vs shRAD51 stalled) = 0.1115. (E) Scatter plot showing the number of RAD51C SIRF signals per cell in control and RNF20-deficient cells in untreated or HU-treated conditions. A total of ≥200 cells were analyzed for each condition from three independent experiments. Black bars represent median values. Mann–Whitney $t$ test, ****$P < 0.0001$. $P$ (shControl stalled vs. shRNF20 stalled) <0.0001. (F) Scatter plot showing the number of RAD51 SIRF signals per cell in control and RNF20-deficient cells in untreated or HU-treated conditions. A total of ≥250 cells were analyzed for each condition from three independent experiments. Black bars represent median values. Mann–Whitney $t$ test, ****$P < 0.0001$. $P$ (shControl stalled vs. shRNF20 stalled) <0.0001. Source data are available online for this figure.

analyzed the status of H2Bub at stalled replication forks in cells expressing RNF20 phosphorylation mutants by the SIRF assay. Interestingly, we found reduced H2Bub SIRF foci in RNF20-depleted cells expressing phosphorylation-deficient mutants of RNF20 compared to WT RNF20 expressing cells (Fig. 6G,H). These results suggest that ATR-mediated RNF20 phosphorylation is required for H2Bub at stalled fork sites to facilitate chromatin remodeling during replication stress responses.

## Discussion

Accurate transmission of genetic information during cell division is crucial for genome maintenance and tumor suppression. Failure to protect the stalled forks from nucleolytic degradation can lead to the accumulation of mutations, gross chromosomal rearrangements and tumorigenesis (Aguilera and Garcia-Muse, 2013; Saxena and Zou, 2022). Many HR factors, including BRCA1/2, RAD51, and RAD51 paralogs, assemble at the stalled fork sites to prevent the degradation of nascent strands and facilitate the restart of stalled forks (Bhattacharya et al, 2022; Rickman and Smogorzewska, 2019; Thakar and Moldovan, 2021; Tye et al, 2021). However, the molecular mechanism by which these proteins localize to the stressed fork sites is largely unclear. RNF20-mediated H2B K120ub is important for the repair of DSBs by HR and NHEJ (Moyal et al, 2011; Nakamura et al, 2011). The data presented here demonstrate an extended role of RNF20-mediated H2Bub at K120 in regulating chromatin dynamics at the stalled fork sites to facilitate the recruitment of BRCA2, RAD51 and RAD51 paralogs to prevent fork degradation and promote fork restart.

Our data shows that RNF20-deficient cells are sensitive to replication stress-inducing agents, and exhibit replication stress markers such as 53BP1 nuclear bodies, micronucleated cells and accumulation of RPA. Notably, we find that RNF20 is enriched at stalled fork sites, and RNF20-mediated H2B K120ub signals were also prominent at these sites. RPA is abundantly present at the stalled fork sites, and we find that RNF20 interacts with RPA, implying that RPA recruits RNF20 to stalled replication sites.

Independent studies have also shown that RNF20/Bre1 interacts with RPA to promote DNA replication and repair of DSBs by HR (Li et al, 2023; Liu et al, 2021). In vitro and In vivo studies from the *Saccharomyces cerevisiae* model system demonstrated that Bre1 interacts with RPA via its LxD/ExD/ExLL motif. Mutations in these residues abolished Bre1 interaction with RPA, resulting in impaired DNA replication, response to replication stress and defective repair of DSBs. Interestingly, residues that are essential for Bre1 interaction with RPA are also conserved in human RNF20, and mutations in these conserved residues abolish RNF20 interactions with RPA (Liu et al, 2021). However, further studies are required to elucidate whether RNF20-dependent replication stress responses in mammalian cells are modulated by RPA in a similar manner as in yeast.

The localization of RNF20 to stalled fork sites is essential for protecting the forks from degradation by MRE11 nuclease and facilitating the replication restart. RAD51, RAD51 paralogs and BRCA2 assemble at the fork sites to prevent its degradation (Liao et al, 2018; Rickman and Smogorzewska, 2019). RAD51 or RAD51C depletion does not affect RNF20 recruitment to fork sites. In contrast, RNF20 knockdown impairs the accumulation of RAD51, RAD51C and BRCA2 at the stalled replication sites, suggesting that RNF20 is upstream and facilitates the loading of RAD51 and RAD51 paralogs to fork sites to protect them from degradation. A previous study showed that BOD1L is required to stabilize RAD51 filaments at the stalled fork sites to prevent fork degradation (Higgs et al, 2015). This occurs via the interaction of BOD1L with SETD1A histone methyl transferase which mediates H3K4 methylation to promote FANCD2-mediated nucleosome remodeling at the stalled fork sites. The FANCD2-dependent histone remodeling subsequently facilitates RAD51 filament stabilization to prevent degradation of stalled forks by the DNA2 nuclease (Higgs et al, 2018). Interestingly, we find that fork protection defects in RNF20-depleted cells can also be rescued by inhibiting DNA2. These data suggest that RNF20-mediated H2B`ub at stalled fork sites is a global response to remodel the chromatin at the stressed fork sites such that BRCA2, RAD51 and RAD51 paralogs localize and protect the stalled forks from MRE11/DNA2-mediated degradation (Fig. 7). However,

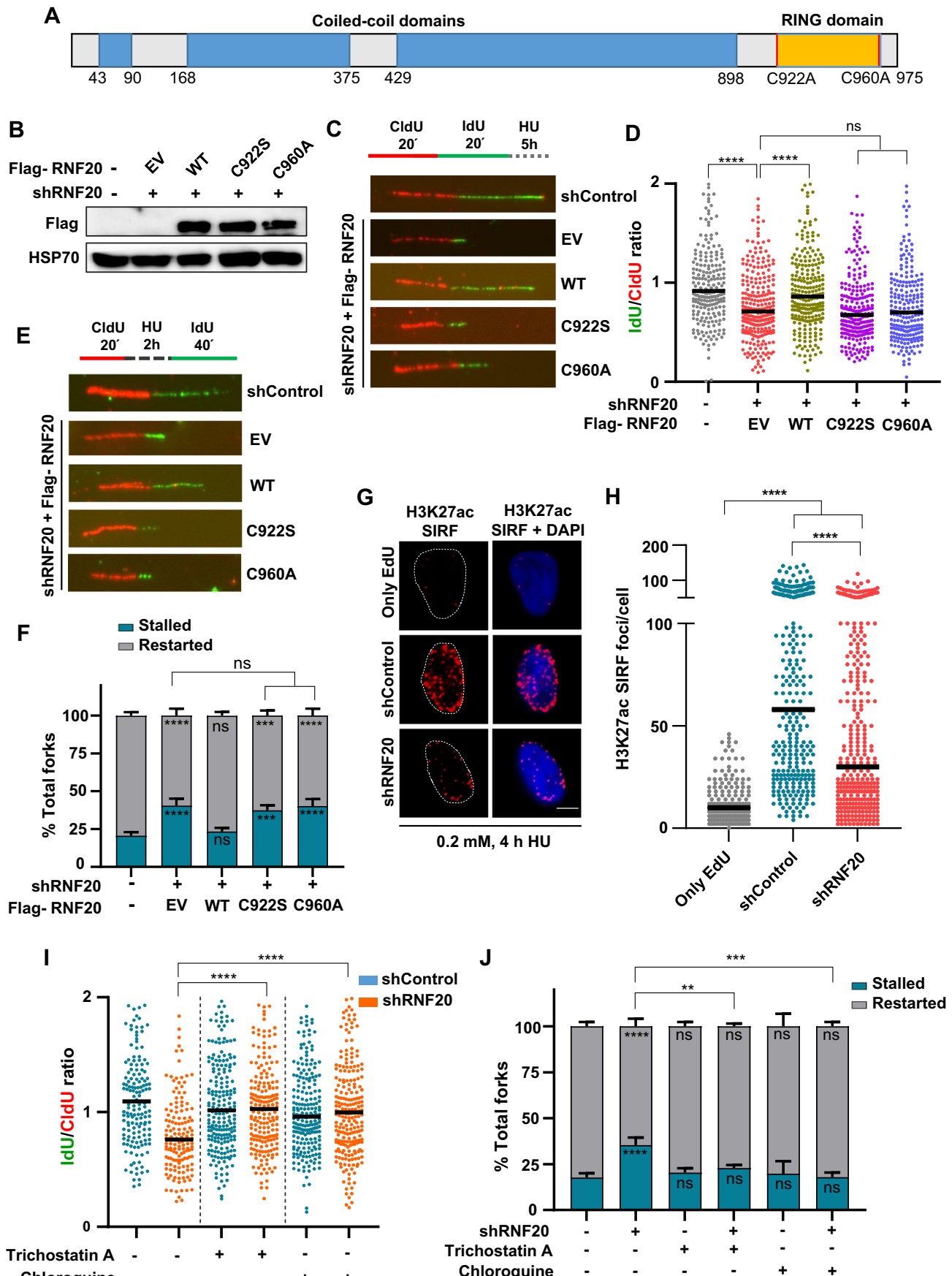

◀ **Figure 5. RNF20 catalytic activity is critical for fork protection and restart.**

(A) Domain architecture of RNF20 E3 ubiquitin ligase. Blue boxes represent coiled-coil domains. Yellow box denotes the RING domain. Sites of point mutation in the RING domain (C922S and C960A) are indicated. (B) Representative western blot showing expression of flag-tagged wild-type (WT), C922S and C960A RNF20. HSP70 serves as the loading control. (C) DNA fiber labeling protocol (top). Representative DNA fibers showing fork protection in the indicated U2OS cells treated with 4 mM HU for 5 h (bottom). (D) Quantification of IdU to CldU tract length ratio in cells as shown in (C). Total of ≥250 fibers were analyzed for each condition from three experimental repeats. Black bars represent median values. Mann–Whitney $t$ test, ****$P < 0.0001$; ns, non-significant. $P$ (shControl vs. shRNF20+ EV; shRNF20 + EV vs. shRNF20 + WT) < 0.0001, $P$ (shRNF20 + EV vs. shRNF20 + C922S) = 0.3105, $P$ (shRNF20 + EV vs. shRNF20 + C960A) = 0.7733. (E) DNA fiber labeling protocol (top). A representative set of DNA fibres showing stalled and restarted forks in the indicated U2OS cells after release from 2 mM, 2 h HU treatment (bottom). (F) Bar graph showing quantification of the percentage of stalled and restarted forks in cells as shown in (E). Stalled and restarted replication forks are shown as percentage of all CldU-labeled tracks. Total of ≥300 fibers were analyzed for each condition from three experimental repeats. Data is represented as mean + SD. Two-way ANOVA, ***$P < 0.001$; ****$P < 0.0001$; ns, non-significant. $P$ (shControl vs. shRNF20 + EV; shControl vs. shRNF20 + C960A) < 0.0001, $P$ (shControl vs. shRNF20 + C922S stalled) = 0.0005, $P$ (shControl vs. shRNF20 + C922S restarted) = 0.0005, $P$ (shRNF20 + EV vs. shRNF20 + C922S) = 0.9845, $P$ (shRNF20 + EV vs. shRNF20 + C960A) > 0.9999. (G) Representative images depicting H3K27ac SIRF signals in control and RNF20-depleted U2OS cells after treatment with 0.2 mM HU for 4 h. Scale bar = 5 μm. Dotted white line indicates the nuclear boundary of the cells. (H) Scatter plot showing the number of H3K27ac SIRF signals per cell in cells as shown in (G). A total of ≥250 cells were analyzed for each condition from three independent experiments. Black bars represent median values. Mann–Whitney test ****$P < 0.0001$. $P$ (shControl vs. shRNF20; Only EdU vs. shControl; Only EdU vs. shRNF20) < 0.0001. (I) Scatter plot showing IdU/CldU ratio in the shControl or shRNF20 transfected U2OS cells treated with TSA (0.2 μM) or chloroquine (20 μg/ml) along with 4 mM HU for 5 h. Total of ≥200 fibers were analyzed for each condition from three experimental repeats. Black bars represent median values. Mann–Whitney $t$ test, ****$P < 0.0001$. $P$ (shRNF20 vs. shRNF20 + TSA; shRNF20 vs. shRNF20 + chloroquine) <0.0001. (J) Bar graph showing percentage of stalled and restarted forks in the indicated U2OS cells treated with HU ± TSA (0.2 μM) or chloroquine (20 μg/ml). Stalled and restarted forks are shown as percentage of all CldU-labeled tracks. Total of ≥200 fibers were analyzed for each condition from three experimental repeats. Data is represented as mean + SD. Two-way ANOVA, **$P < 0.01$, ***$P < 0.001$, ****$P < 0.0001$; ns, non-significant. $P$ (shControl UT vs. shRNF20 UT) < 0.0001, $P$ (shRNF20 UT vs. shRNF TSA stalled) = 0.0058, $P$ (shRNF20 UT vs. shRNF chloroquine stalled) = 0.0001, $P$ (shRNF20 UT vs. shRNF TSA restarted) = 0.0058, $P$ (shRNF20 UT vs. shRNF chloroquine restarted) = 0.0001. Source data are available online for this figure.

further studies are required to understand whether SETD1A-mediated H3K4 methylation is dependent on RNF20-mediated H2Bub.

The expression of RNF20 RING domain mutants resulted in defective fork protection and restart, implying that RNF20-mediated H2Bub is critical for fork stabilization and recovery. In response to DSBs, ATM kinase phosphorylates RNF20 at S172 and S553, and this phosphorylation is essential for RNF20-mediated H2Bub, which in turn facilitates the recruitment of RAD51 and promotes the repair of DSBs by HR (Moyal et al, 2011). In response to replication stress, ATR kinase activates the replication checkpoint, slows down the DNA replication, and facilitates fork protection and its restart (Saldivar et al, 2017; Simoneau and Zou, 2021). ATR regulates replication stress responses by targeting a large number of proteins (Matsuoka et al, 2007). Indeed, we find that RNF20 undergoes phosphorylation in an ATR-dependent manner in response to replication stress. Strikingly, we find that RNF20-mediated fork protection and replication restart are dependent on the phosphorylation of RNF20. Interestingly, H2B K120ub signals were significantly reduced in cells expressing RNF20 phosphomutants, suggesting that RNF20 phosphorylation regulates the catalytic activity of RNF20.

How does RNF20-mediated H2Bub regulate replication stress response? Various studies indicate that H2Bub is required for chromatin decompaction (Kato and Komatsu, 2015). The addition of ubiquitin on H2B K120 adds a significant bulk to the nucleosome, introducing direct structural changes in chromatin, leading to its decompaction (Fierz et al, 2011). Moreover, RNF20-mediated H2Bub facilitates the recruitment of SNF2H, which belongs to the ISWI family of ATP-dependent chromatin remodelers to the sites of DSBs (Nakamura et al, 2011; Oliveira et al, 2014; Smeenk et al, 2013; Toiber et al, 2013). The repair defects in RNF20/SNF2H depleted cells were rescued by chromatin-relaxing agents (Nakamura et al, 2011). Using LacR/LacO heterochromatin relaxation assay, RNF20 has been shown to induce chromatin relaxation in an SNF2H-dependent manner to promote DSB repair in heterochromatin (Klement et al, 2014). These data suggest that RNF20-mediated H2Bub facilitates chromatin relaxation during transcriptional elongation and DSB repair (Klement et al, 2014; Pavri et al, 2006). Notably, the chromatin relaxing agents (chloroquine and TSA) rescued replication fork protection and restart defects in RNF20-deficient cells in our assays, providing evidence for RNF20-mediated chromatin decompaction at stalled fork sites, which facilitates the recruitment of RAD51 and RAD51 paralogs to stalled forks to promote their protection and restart (Fig. 7). However, further studies are required to understand whether RNF20-mediated H2Bub is essential for accumulating SNF2H or other factors at the stalled fork sites to promote chromatin decompaction. In addition to its enzymatic activity, RNF20 has been shown to serve as an adaptor to recruit BRCA1 to damage sites via TRAIP and RAP80 (Soo Lee et al, 2016). Whether RNF20 similarly recruits BRCA1 or other HR factors to stalled fork sites by its physical interaction needs further investigation.

A previous study showed that H3K9me3 modification occurs near DSB ends, which promotes transient repressive chromatin to stabilize the damaged chromatin. This modification is required to recruit Tip60 histone acetyltransferase which activates ATM and promotes H4 acetylation to facilitate chromatin decompaction (Ayrapetov et al, 2014). The ATM-mediated phosphorylation of kap1 also promotes the release of the kap1/HP1/suv39h1 repressive complex. These dynamic responses at the damaged chromatin facilitate the recruitment of repair factors and promote the efficient repair of DSBs (Ayrapetov et al, 2014; Price and D'Andrea, 2013). Notably, a previous study showed that G9a histone methyl transferase promotes H3K9me3 at the stalled fork sites, and this modification facilitates chromatin compaction transiently to stabilize the stalled forks from degradation by nucleases (Gaggioli et al, 2023). Interestingly, defects in this modification also resulted in impaired localization of fork protection factors such as BRCA1, FANCD2 and RAD51 (Gaggioli et al, 2023). However, our data shows that RNF20-mediated chromatin decompaction facilitates

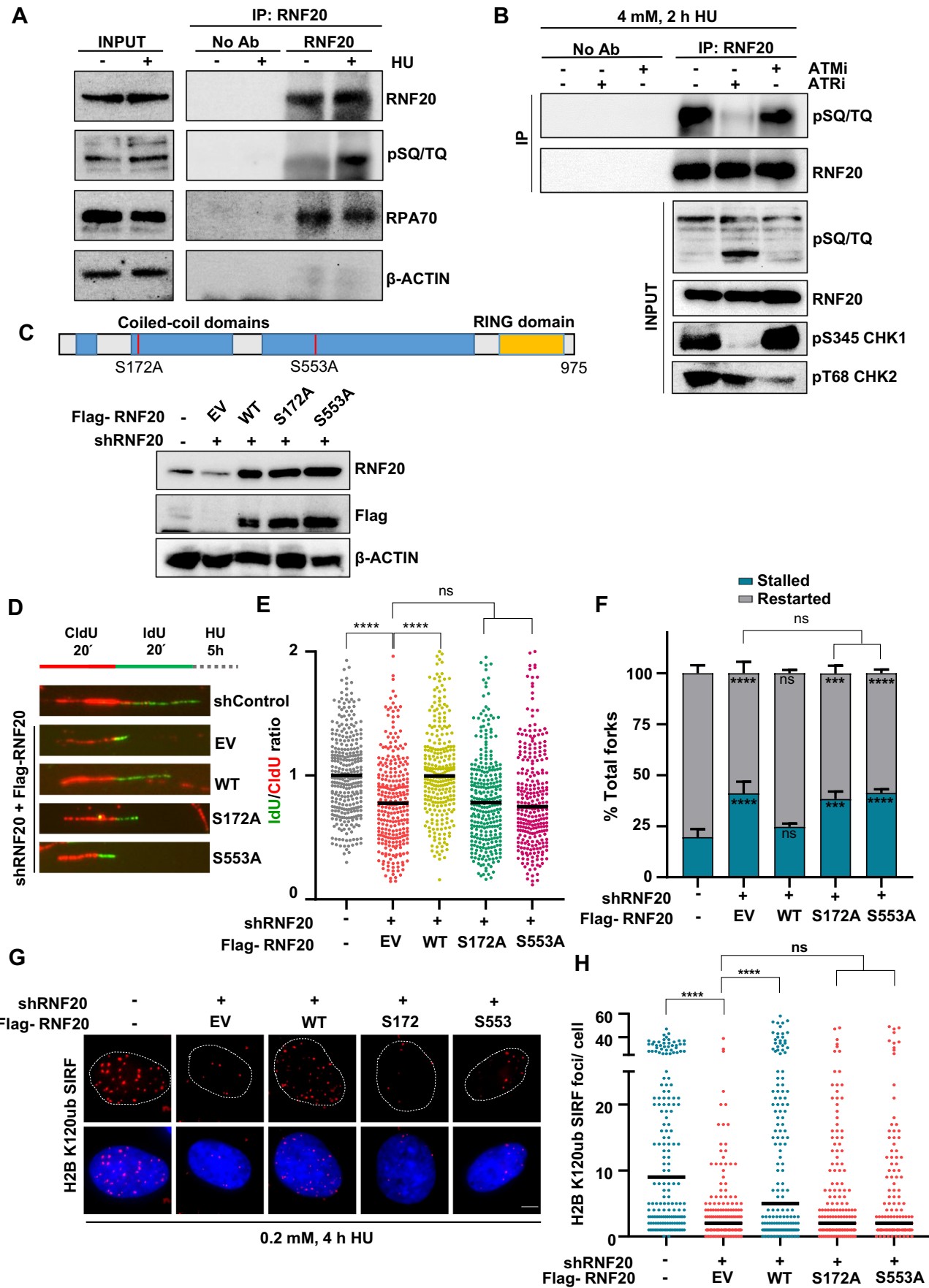

**Figure 6. Phosphorylation of RNF20 by ATR is essential for efficient protection and restart of stalled forks.**

(**A**) Representative RNF20 co-immunoprecipitation blot (Co-IP) showing phosphorylated RNF20 (as detected by an anti-phospho-SQ/TQ antibody) in mock or HU-treated HeLa cells. RPA70 co-immunoprecipitated along with RNF20 in both UT and HU-treated conditions. β-ACTIN was taken as a negative control in the Co-IP reaction. (**B**) Representative western blot of RNF20 co-IP showing phosphorylated RNF20 (as detected by an anti-phospho-SQ/TQ antibody) in HeLa cells treated with 4 mM HU for 2 h along with ATM or ATR kinase inhibitor. (**C**) Domain architecture of RNF20 ubiquitin ligase. RNF20 phosphorylation sites and their mutants (S172A and S553A) are indicated (top). Representative immunoblot showing expression of flag-tagged wild-type (WT), S172A and S553A RNF20 constructs. Actin serves as the loading control (bottom). (**D**) DNA fiber labeling protocol (top). Representative DNA fibers showing the extent of fork protection in the indicated U2OS cells (bottom). (**E**) Quantification of IdU to CldU tract length ratio in cells as shown in (**D**). Total of ≥250 fibers were analyzed for each condition from three independent experiments. Black bars represent median values. Mann–Whitney t test, ****P < 0.0001; ns, non-significant. P (shControl vs. shRNF20 + EV; shRNF20 + EV vs. shRNF20 + WT) < 0.0001, P (shRNF20 + EV vs. shRNF20 + S172A) = 0.7998, P (shRNF20 + EV vs. shRNF20 + S553A) = 0.8109. (**F**) Quantification of percentage of stalled and restarted forks in the indicated U2OS cells. Stalled and restarted replication forks are shown in bar graph as percentage of all CldU-labeled tracks. Total of ≥200 fibers were analyzed for each condition from three independent experiments. Data is represented as mean ± SD. Two-way ANOVA, ***P < 0.001; ****P < 0.0001; ns, non-significant. P (shControl vs. shRNF20 + EV, shControl vs. shRNF20 + S553A) < 0.0001, P (shControl stalled vs. shRNF20 + S172A stalled) = 0.0002, P (shControl restarted vs. shRNF20 + S172A restarted) = 0.0002, P (sRNF20 + EV vs. shRNF20 + S172A) = 0.9930, P (sRNF20 + EV vs. shRNF20 + S553A) > 0.9999. (**G**) Representative images depicting H2B K120ub SIRF signals in U2OS cells expressing either WT or S172A/S553A RNF20 after endogenous RNF20 depletion and treated with HU (0.2 mM, 4 h). Scale bar = 5 µm. Dotted white lines represent nuclear boundary of the cells. (**H**) Quantification of H2B K120ub SIRF signals in cells as shown in (**G**). Total of ≥250 cells were analyzed for each condition. Black bars represent median values. Mann–Whitney t test, ****P < 0.0001; ns, non-significant. P (shControl vs. shRNF20 + EV; shRNF20 + EV vs. shRNF20 + WT) < 0.0001, P (shRNF20 + EV vs. shRNF20 + S172A) = 0.3403, P (shRNF20 + EV vs. shRNF20 + S553A) = 0.6663. Source data are available online for this figure.

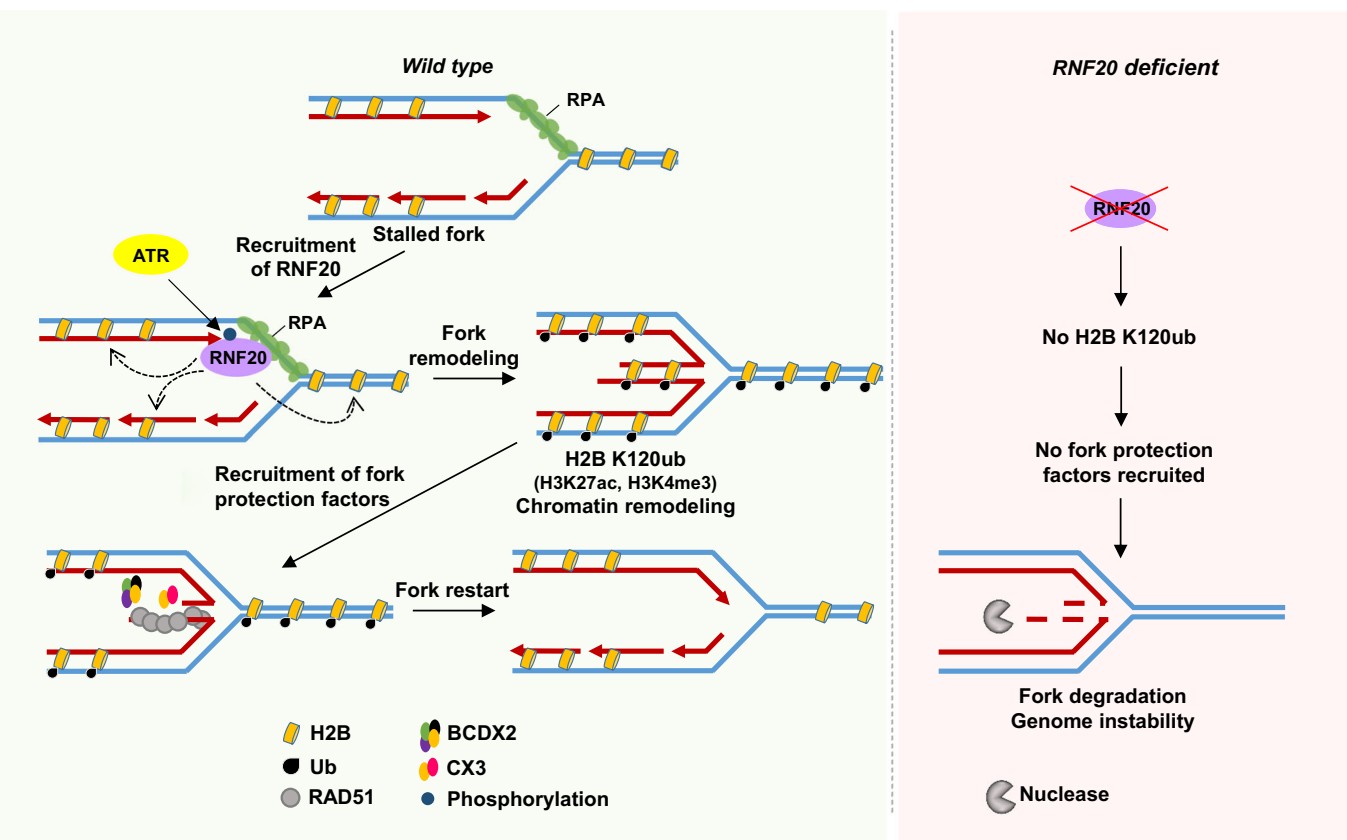

**Figure 7. A model for RNF20-mediated H2Bub regulating chromatin remodeling for fork protection and restart.**

RPA recruits RNF20 to the stalled fork sites. ATR phosphorylates RNF20 and promotes RNF20-mediated H2B K120ub at the stalled replication sites. H2Bub facilitates chromatin remodeling (decompaction) to promote recruitment of RAD51 and RAD51 paralogs at the sites of stalled forks to safeguard the replicating genomes.

the recruitment of RAD51 and RAD51 paralogs to the stalled fork sites for stabilization (Fig. 7). The co-existence of H3K9me3 and H2B K120ub chromatin marks at the stalled fork sites may be temporally and dynamically regulated to protect the stalled forks and mediate subsequent restart. Nonetheless, further studies are required to understand how the differential chromatin modifications regulate the chromatin dynamics to deal with replication stress responses.

# Methods

### Reagents and tools table

| Reagent/resource | Reference or source | Identifier or catalog number |
|---|---|---|
| **Experimental models: cell lines** | | |
| U2OS | ATCC | HTB-96; RRID:CVCL_0042 |
| HeLa Kyoto | Sachin Kotak Lab | RRID: CVCL_1922 |
| **Recombinant DNA** | | |
| pcDNA3β-Flag WT RNF20 | This paper | N/A |
| pcDNA3β-Flag C922S RNF20 | This paper | N/A |
| pcDNA3β-Flag C960A RNF20 | This paper | N/A |
| pcDNA3β-Flag S172A RNF20 | This paper | N/A |
| pcDNA3β-Flag S552A RNF20 | This paper | N/A |
| **Antibodies** | | |
| Anti-RNF20 (for WB and PLA) | Abcam | Cat# ab3269 |
| Anti- β-ACTIN (for WB) | Santa Cruz | Cat# sc-81178 |
| Anti-Cyclin A (for IF) | Santa Cruz | Cat# sc-751 |
| Anti-53BP1 (for IF) | Merck | Cat# 05-726 |
| Anti-RPA70 (for IF and WB) | Abcam | Cat# ab79398; RRID: AB_1603759 |
| Anti-γH2AX (for IF) | BD Biosciences | Cat# 560443; RRID: AB_1645592 |
| Anti-RAD51(for PLA) | Abcam | Cat# ab-176458; RRID: AB_2665405 |
| Anti-RAD51C (for PLA) | Abcam | Cat# ab-72063; RRID: AB_2177279 |
| Anti-PCNA (for WB) | Santa Cruz | Cat# sc-56 |
| Anti-H3 (for WB) | Gift from Dr. Srimonta Gayen lab | N/A |
| Anti-H2 (for WB) | Abcam | Cat# ab1790 |
| Anti-H2B K120ub (for PLA) | Merck | Cat# 05-1312I |
| Rat anti-BrdU (For fiber) | Abcam | Cat# ab6326; RRID: AB_305426 |
| Donkey Anti-Rat Alexa Fluor 594 (For fiber) | Abcam | Cat# ab150156; RRID: AB_2890252 |
| Purified Mouse Anti-BrdU (For fiber) | BD Biosciences | Cat# 347580; RRID: AB_400326 |
| Rabbit Anti-Mouse IgG H&L Alexa Fluor 488 (For fiber) | Abcam | Cat# ab150125 |
| Anti-Flag (for WB and PLA) | Sigma-Aldrich | Cat# F1804; RRID: AB_262044 |
| Anti-HSP70 (for WB) | Santa Cruz | Cat# sc32239 |
| Anti-H3K27ac (for PLA) | Abcam | Cat# ab302877 |
| Phospho-ATM/ATR Substrate Motif [(pS/pT) QG] MultiMab® Rabbit mAb mix (for WB) | Cell Signaling Technology | Cat# 6966S |
| Anti-CHK1 pS345 (for WB) | Cell Signaling Technology | Cat# 2348S |
| Anti-CHK2 pT68 (for WB) | Cell Signaling Technology | Cat# 2197 |
| Anti-FANCD2 (for IF) | Santa Cruz | Cat# sc20022 |
| Anti-RPA32 pS4/8 (for WB) | Bethyl | Cat# A300-245A |
| Anti-SMARCAL1 (for WB) | Abcam | Cat# ab-154226 |
| Anti-ZRANB3 (for WB) | Abcam | Cat# ab109595; RRID: AB_10866685 |

| Reagent/resource | Reference or source | Identifier or catalog number |
|---|---|---|
| Anti-HLTF (for WB) | Santa Cruz | Cat# sc-398357 |
| Anti-RAD51C (for WB) | Santa Cruz | Cat# sc-56214; RRID: AB_2238197 |
| Anti-XRCC2 (for WB) | Santa Cruz | Cat# sc-365854; RRID: AB_10846464 |
| Anti-XRCC3 (for WB) | Santa Cruz | Cat# sc-271714; RRID: AB_10708416 |
| Anti-BRCA2 (for PLA) | Santa Cruz | Cat# sc293185 |
| Anti-H3K4me3 (for PLA) | Abcam | Cat# ab8580 |
| Anti-ORC2 (for WB) | Santa Cruz | Cat# sc-398410 |
| Mouse anti-rabbit IgG-HRP | Santa Cruz | Cat# sc-2357 |
| m-IgGk BP-HRP | Santa Cruz | Cat# sc-516102 |
| **Oligonucleotides and other sequence-based reagents** | | |
| shRNF20 #1: (5′-GGGGTGAG AGCTGGAATCTCTGC-3′) | Sigma-Aldrich | N/A |
| shRNF20 #2: (5′-GAAGGCA GCTGTTGAAGATTC-3′) | Sigma-Aldrich | N/A |
| shRAD51C (5′-CACCTTCTGTTC AGCACTAGA-3′) | Sigma-Aldrich | N/A |
| shXRCC2 (5′-TTGCAACGACAC AAACTATAA-3′) | Sigma-Aldrich | N/A |
| shXRCC3 (5′-GAATTATTGCTG CAATTAA-3′) | Sigma-Aldrich | N/A |
| shSMARCAL1 (5′-GCTTTGACC TTCTTAGCAAT-3′) | Sigma-Aldrich | N/A |
| shZRANB3 (5′-TGGTGTGTGT CAGCTCTGT-3′) | Sigma-Aldrich | N/A |
| shHLTF (5′-GGAATATAATG TTAACGAT-3′) | Sigma-Aldrich | N/A |
| RNF20 primers (refer to Table EV2) | Sigma-Aldrich | N/A |
| **Chemicals, enzymes and other reagents** | | |
| Hydroxyurea | Sigma-Aldrich | Cat# H8627 |
| Mirin | Sigma-Aldrich | Cat# M9948 |
| C5 | Medchem Express | Cat# HY-128729 |
| DAPI | Sigma-Aldrich | Cat# D8417 |
| 5-Chloro-2-deoxyuridine | Sigma-Aldrich | Cat# C6891 |
| 5-Iodo-2-deoxyuridine | Sigma-Aldrich | Cat# I7125 |
| cOmpleteTM, Mini Protease Inhibitor Cocktail | Roche | Cat# 11836153001 |
| PhosSTOP | Roche | Cat# 4906837001 |
| Thiazolyl Blue Tetrazolium Bromide | Sigma-Aldrich | Cat# M2128 |
| KaryoMAXTM ColcemidTM Solution | Thermo Fisher Scientific | Cat# 15212012 |
| Agarose, low gelling temperature | Sigma-Aldrich | Cat# A9414 |
| Protein-G Sepharose beads | Cytiva | Cat# GE17-0618-01 |
| 5-ethynyl-2-deoxyuridine | Thermo Fisher Scientific | A10044 |
| L-Ascorbic acid | Sigma-Aldrich | Cat# A92902 |
| Mowiol® 4-88 | Merck | Cat# 81381 |
| Dynabeads™ MyOne™ Streptavidin C1 | Invitrogen | Cat# 65002 |
| Aphidicolin | Sigma-Aldrich | Cat# A0781 |
| Immobilon Western Chemiluminescent HRP Substrate | Millipore | Cat# WBKLS0500 |
| Duolink® In Situ PLA® Probe Anti-Rabbit PLUS | Merck | Cat# DUO92002 |
| Duolink® In Situ PLA® Probe Anti-Mouse MINUS | Merck | Cat# DUO92004 |

| Reagent/resource | Reference or source | Identifier or catalog number |
|---|---|---|
| Duolink® In Situ Detection Reagents Red | Merck | Cat# DUO92008 |
| BamHI-HF | New England Biolabs | Cat# R3136 |
| HindIII-HF | New England Biolabs | Cat# R3104 |
| Q5® High-Fidelity DNA Polymerase | New England Biolabs | Cat# M0491 |
| VE821 (ATR inhibitor) | Sigma-Aldrich | Cat# SML1415 |
| KU55933 | Sigma-Aldrich | Cat# SML1109 |
| **Software** | | |
| ImageJ | ImageJ | https://imagej.net/software/fiji/downloads |
| GraphPad Prism 9 | GraphPad software | https://www.graphpad.com/ |
| **Other** | | |
| ChemiDoc MP Imaging System | Bio-Rad | N/A |
| Zeiss Axio Observer | Zeiss | N/A |

## Cell lines and culture conditions

Human cell lines U2OS and HeLa were grown in Dulbecco's Modified Eagle Medium (DMEM) supplemented with 10% FBS, 1% penicillin/streptomycin (Sigma-Aldrich) and 1% Glutamax (Gibco) at 37 °C in a humidified air chamber containing 5% $CO_2$.

## Plasmids and transfections

The plasmid encoding full-length WT human RNF20 was purchased from Origene and cloned into pcDNA3β expression vector by PCR amplification. All RING domain and phosphorylation site mutants of RNF20 were generated by site-directed mutagenesis and cloned into pcDNA3β vector. The sequence of the primers used for mutagenesis has been mentioned in Table EV2. RNF20 UTR-specific shRNA was generated for this paper, and gene-specific shRNA was generated using the reported siRNA sequence and cloned into a pRS shRNA vector. All shRNA sequences used are listed in Table EV1. All plasmid transfections for transient depletion/expression were performed using a Bio-Rad gene pulsar X cell (260 V and 1050 μF). Fresh media was added to the cells 6–8 h after transfection. Cells were processed for indicated treatments/experiments 24–30 h after transfection.

## Immunoblotting

Immunoblotting was performed as previously described (Nath and Nagaraju, 2020). Briefly, cells were harvested and lysed in RIPA lysis buffer (50 mM Tris-HCl pH 7.5, 1% NP-40, 0.5% sodium deoxycholate, 0.1% SDS, 150 mM NaCl, 2 mM EDTA, and 50 mM sodium fluoride) supplemented with cOmplete mini protease inhibitor cocktail (Roche). Protein estimation was done by standard Bradford assay. 30–50 μg of proteins were resolved on SDS-PAGE gel and transferred onto PVDF membranes (Millipore) by semi-dry transfer method (Bio-Rad Trans-Blot SD). Membranes were blocked using 5% skim-milk (Hi-media) in TBST (50 mM Tris-

HCl, pH 8.0, 150 mM NaCl, 0.1% Tween-20) and incubated with primary antibody overnight (O/N) at 4 °C, followed by HRP-conjugated secondary antibody incubation for 1 h at room temperature (RT). After TBST washes, membranes were developed with chemiluminescent HRP substrate (Millipore) and imaged using Chemidoc (Bio-Rad Chemidoc Imaging System). The following primary antibodies were used in this study for western blotting: rabbit anti-RNF20 (1:2000, Abcam), mouse anti-β-ACTIN (1:2000, Santa Cruz (SC)), rabbit anti-SMARCAL1 (1:500, Abcam), rabbit anti-ZRANB3 (1:500, Abcam), mouse anti-HLTF (1:500, SC), mouse anti-RAD51C (1:250, SC), mouse anti-XRCC2 (1:250, SC), mouse anti-XRCC3 (1:200, SC), mouse anti-Flag (1:1000, Sigma), mouse anti-MCM3 (1:2000, SC), mouse anti-RPA70 (1:500, SC), and mouse anti-HSP70 (1:1000, SC).

## Immunofluorescence

For native BrdU staining, cells were incubated with 25 μM BrdU for 48 h, washed and treated with 4 mM HU for 2 h. For scoring FANCD2 foci in mitotic cells, cells were treated with 0.4 μM APH for 22 h with addition of RO3306 for the last 6 h. Cells were released in fresh media for 30 min followed by mitotic shake-off and plated in poly-L-lysine coated coverslips. For the remaining experiments, exponentially growing cells were seeded onto coverslips 8 h after transfection, then treated (or mock-treated) with HU as indicated. After treatment, the cells were washed with PBS, pre-extracted with 0.5% Triton X-100 for 90 s on ice and fixed in 4% formaldehyde for 10 min at RT. After three PBS washes, coverslips were blocked in blocking buffer (0.5% BSA and 0.5% Triton X-100 in PBS) for 30 min. The coverslips were incubated with the indicated primary antibodies for 2 h at RT. After a wash with blocking buffer, the coverslips were incubated with respective FITC/TRITC-conjugated secondary antibodies for 1 h at RT and then stained with DAPI (1 μg/ml; Sigma-Aldrich) for 10 min before mounting onto slides with Mowiol 4–88 (Sigma). Images were acquired using an Apotome microscope (Zeiss Axio Observer) and processed using ImageJ software. The following antibodies were used for performing immunofluorescence experiments in this study: rabbit anti-53BP1 (1:500, Novus Biologicals), mouse anti-Cyclin A (1:200, SC), rabbit anti-RPA70 (1:2000, Abcam), mouse anti-H2AX (pS139) (1:1000, BD Biosciences), rabbit anti-Phospho RPA32(pS4/8) (1:2000, Bethyl Laboratories), rat anti-BrdU (1:500, Abcam), mouse anti-Flag (1:1000, Sigma), rabbit anti-RAD51 (1:500, Abcam) and mouse anti-RAD51C (1:100, SC).

## Cell survival

5000 cells per well were seeded in a 24-well plate for each condition. After treatment (or mock treatment) with HU (continuous) and APH (48 h), cells were allowed to grow for 5–7 days. Later, cell survival was measured by MTT (0.3 mg/ml; Sigma-Aldrich) assay using a microplate reader (VersaMaxROM version 3.13). Percent cell survival was calculated as treated cells/untreated cells*100.

## Quantitative in-situ analysis of protein interactions at DNA replication forks (SIRF)

Experiments were performed as described previously (Roy et al, 2018). Exponentially growing cells were plated in 12 well plates containing coverslips the day before the experiment. Cells were incubated with 125 μM EdU for 8 min followed by HU (200 μM) or

thymidine (100 μM) for 4 h and fixed with 2% PFA in PBS for 15 min at RT. Next, cells were permeabilized with 0.25% Triton X-100 in PBS for 15 min at RT. After 2x PBS washes, the Click reaction cocktail (2 mM copper sulfate, 10 μM biotin-azide, and 100 mM sodium ascorbate in PBS) was freshly prepared and added to the coverslips in a humidified chamber for 1 h at RT. After the click reaction, slides were washed with PBS and blocked with blocking buffer (10% goat serum and 0.1% Triton X-100 in PBS) for 1 h at RT followed by incubation with primary antibodies at 4 °C, O/N. Mouse or rabbit anti-biotin antibody was used in conjunction with the respective antibody for the protein of interest. On the next day, PLA reactions were performed according to manufacturer's protocol (Duolink Proximity Ligation assay, Merck). Briefly, coverslips were incubated with anti-mouse minus and anti-rabbit plus probes in a humidified chamber for 1 h at 37 °C, followed by ligation in a humidified chamber for 30 min at 37 °C and finally, amplification of the annealed probes in a humidified chamber for 100 min at 37 °C. Coverslips were stained with DAPI for 10 min before mounting onto slides with mounting medium (Mowiol 4–88, Sigma). Cells were imaged using an Apotome microscope (Zeiss Axio Observer), and PLA foci were quantified using ImageJ software. Antibodies used for SIRF assay in this study include: rabbit anti-biotin (1:1000, CST), mouse anti-biotin (1:1500, Invitrogen), rabbit anti-RNF20 (1:500, Abcam), rabbit anti-RAD51 (1:250, Abcam), rabbit anti-RAD51C (1:100, Abcam) and mouse anti-ubiquityl histone H2B (1:250, Merck).

## DNA fiber assay

Cells were plated in a six-well plate after transfection, and after 24 h, cells were sequentially pulse-labeled with 25 μM CldU (Sigma) and 250 μM IdU (Sigma) followed by 4 mM HU treatment for 5 h for fork protection assay. Alternatively, cells were pulse-labeled with 25 μM CldU and treated with 2 mM HU for 2 h to induce fork stalling. This was followed by recovery into fresh media containing 250 μM IdU for fork restart assay. Next, the cells were incubated in ice-cold PBS on ice for 10 mins, harvested, counted and re-suspended in 250 μl PBS. In total, 3 μl of the cell mixture was mixed with 7 μl of lysis buffer on glass slides (ThermoScientific superfrost) and allowed to stand for 7 min. Slides were inclined at a 45° angle to spread the suspension and fixed in methanol:acetic acid (3:1) solution at 4 °C, O/N. The following day, DNA was denatured by incubating in 2.5 M HCl for 1 h and blocked with 2% BSA in 0.1% PBST solution (1× PBS and 0.1% Tween-20). Next, the slides were incubated with primary antibodies for 2.5 h and secondary antibodies for 1 h at RT. Coverslips were mounted on the slides with a Mowiol mounting medium (Sigma) and visualized using an Apotome microscope (Zeiss Axio observer). Fiber length was measured using ImageJ software from 3 independent experiments and P values were calculated using Prism software. Antibodies used for performing DNA fiber studies include: rat anti-BrdU for CldU (1:500, Abcam), mouse anti-BrdU for IdU (1:250, BD Biosciences), rabbit anti-mouse IgG (Alexa Fluor 488) (1:500, Abcam) and donkey anti-rat IgG (Alexa Fluor 594) (1:500, Abcam).

## Co-immunoprecipitation

After indicated treatments cells, 5 million cells per condition were harvested and lysed in RIPA lysis buffer (without SDS) containing cOmplete mini protease inhibitor and PhosSTOP phosphatase inhibitor cocktail (Roche). In all, 2 mg protein from each sample was either incubated with no antibody or rabbit polyclonal anti-RNF20 antibody O/N at 4 °C under rotatory agitation. The following day, protein G beads were washed multiple times in lysis buffer and incubated with cell lysate and antibody mixture for 4 h at 4 °C under rotatory agitation. Following incubation, the beads were washed 3 times in lysis buffer and protein was eluted by boiling the beads in 2x Laemmli buffer for 15 min. Western blotting (WB) was performed as described in the previous section. Antibodies used for performing immunoprecipitation and western blotting include: rabbit anti-RNF20 (1.5 μg for IP; 1:2000 for WB, Abcam), rabbit anti-(pS/pT)Q antibody (1:500, CST), mouse anti-RPA70 (1:1000, SC), rabbit anti-CHK1 pS345 (1:1000, CST), rabbit anti-CHK2 pT68 (1:1000, CST) and mouse anti-β-ACTIN (1:2000, SC).

## Comet assay

Frosted glass slides (Bluestar) were coated with 1% agarose at least 24 h before the experiment. Cells were treated with HU, as mentioned and harvested in PBS. After centrifugation, 30 μl of cell suspension was mixed with 270 μl of 0.5% low melting point agarose and 100 μl of this mixture was spread onto pre-coated slides. Slides were incubated in chilled lysis buffer (2.5 M NaCl, 0.1 M EDTA, 10 mM Tris-HCl pH 8, 1% Triton X-100 and 10% DMSO) O/N at 4 °C. The following day, the slides were washed in electrophoresis buffer (300 mM sodium hydroxide and 1 mM EDTA, pH >13) and transferred to an electrophoresis tank filled with chilled electrophoresis buffer. The electrophoresis was performed at 1 V/cm for 20 min at RT. Slides were then washed with PBS, fixed in methanol for 5 mins at RT, washed with double-distilled $H_2O$ and transferred to 70% ethanol, followed by 100% ethanol for 15 min each at RT. Slides were then air-dried and stained with propidium iodide (2 μg/ml in Milli-Q). Images were acquired using an Apotome microscope (Zeiss Axio Observer). Comet tail moment was measured using OpenComet plug-in in ImageJ software.

## Metaphase spreads

shRNA transfected cells were treated with 2 mM HU for 4 h along with 0.1 μg/mL colcemid (KaryoMAX, Gibco) 30 h post-transfection. Cells were then harvested and re-suspended in hypotonic solution (75 mM KCl in Milli-Q) and incubated in 37 °C waterbath for 12 min. After centrifugation at 160x g for 10 min, cells were fixed in 5 ml of methanol: acetic acid (3:1) fixative. Fixed cells were lysed by dropping 100 μl of the cell suspension onto chilled slides, followed by incubation on a steaming water beaker (70–80 °C) for 90 s. Slides were then air-dried and stained with Geimsa (Sigma) stain for 30 min at RT. Excess stain was washed off, and the slides were air-dried. At least 50 metaphase spreads were scored from three independent experiments using Olympus BX53 microscope.

## iPOND (Isolation of proteins on nascent DNA)

iPOND was performed as described previously (Dungrawala et al, 2015; Sirbu et al, 2012). Briefly, one hundred million exponentially growing HeLa cells were used per condition and treated as

described. For capturing proteins associated with progressing forks, cells were pulse-labelled with 10 µM EdU for 20 min. For capturing proteins at stalled forks, 3 mM HU was added for 2 h following EdU-labeling. For thymidine chase, cells were incubated with 10 µM thymidine for 2 h following EdU-labeling. Cells were crosslinked with 1% formaldehyde for 20 min, quenched with 0.125 M glycine and washed three times with PBS. Cells were then permeabilized with 0.25% Triton X-100 in PBS for 30 min at room temperature and washed twice with 0.5% BSA in PBS. Next, EdU was conjugated to biotin by click reaction where cells were incubated in click reaction buffer (10 mM biotin azide, 10 mM sodium ascorbate and 2 mM $CuSO_4$ in PBS) for 2 h at room temperature. Cells were washed twice with 0.5% BSA solution and re-suspended in lysis buffer (50 mM Tris-HCl, pH 8, and 1% SDS) supplemented with protease inhibitors. Chromatin was extracted by sonication followed by centrifugation at 16,100 rcf for 10 min. Supernatants were diluted with 1:1 PBS (vol/vol) containing protease inhibitors and incubated overnight with streptavidin-conjugated Dynabeads. Beads were washed once with lysis buffer, once with low salt buffer (1% Triton, 20 mM Tris pH 8, 2 mM EDTA, 150 mM NaCl), once with high salt buffer (1% Triton, 20 mM Tris pH 8, 2 mM EDTA, 500 mM NaCl), and twice with lysis buffer. Captured proteins were eluted by boiling beads for 30 min at 95 °C in 2× Laemmli buffer and proteins were analysed by western blot analysis.

## IdU-co-immunoprecipitation of proteins present at the replication forks

shControl or shRNF20 transfected HeLa cells ($10 \times 10^6$) were labeled with 100 µM IdU for 30 min followed by treatment with 3 mM HU for 2 h. Cells were then crosslinked in 1% paraformaldehyde for 20 min, quenched with 0.125 M glycine for 5 min and washed 3 times with PBS. The cytosolic protein fraction was removed by incubation in cytosolic buffer (10 mM HEPES, pH 7, 50 mM NaCl, 0.3 M sucrose, 0.5% Triton X-100) supplemented with protease inhibitor for 15 min on ice and centrifuged at $1500 \times g$ for 5 min. The soluble nuclear fraction was removed by incubation with nuclear buffer (10 mM HEPES, pH 7, 200 mM NaCl, 1 mM EDTA, 0.5% NP-40 and protease inhibitor cocktail) for 10 min on ice and then centrifuged at 13,000 rpm for 2 min. The pellets were re-suspended in lysis buffer (10 mM HEPES, pH 7, 500 mM NaCl, 1 mM EDTA, 1% NP-40 and protease inhibitor cocktail), sonicated at low amplitude for chromatin extraction and centrifuged for 1 min at 13,000 rpm. A total of 250 µg protein was used for IP with 5 µg anti-IdU antibody and 20 µl of Protein G-agarose (GE Healthcare). The IP reaction was washed twice with nuclear buffer and twice with washing buffer (10 mM HEPES, 0.1 mM EDTA and protease inhibitor cocktail), incubated in 2×Laemmli buffer for 30 min at 90 °C, and proteins were analysed by western blotting.

## Quantification and statistical analysis

All experiments reported were independently replicated at least three times, unless mentioned otherwise. Data represent mean ± SD/SEM from at least three independent experiments. The statistical analysis of all experiments was performed using GraphPad Prism Version 9. The statistical significance of DNA fiber experiments, unpaired immunofluorescence experiments,

SIRF experiments, comet assay and metaphase spread experiments were determined by Mann–Whitney $t$ test and Kruskal–Wallis test for multiple comparisons. The statistical significance of grouped immunofluorescence experiments was determined by two-way ANOVA. Significance is indicated by asterisk (*$P < 0.05$; **$P < 0.01$; ***$P < 0.001$; ****$P < 0.0001$. n.s., non-significant.) and p < 0.05 was considered statistically significant.

## Data availability

No primary database has been generated for this study.

The source data of this paper are collected in the following database record: biostudies:S-SCDT-10_1038-S44319-025-00497-3.

## Peer review information

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

## Acknowledgements

We thank Nitika Taneja, Arnab Ray Choudhuri, Wolf Heyer, and Kumar Somyajit for their insightful suggestions. We thank Benu Brata Das, Kundan Sengupta and Srimonta Gayen for providing reagents. We thank members of the GN lab for their useful discussions and proofreading the manuscript. Funding by Department of Science and Technology (EMR/2015/001720; CRG/2022/003533); Department of Atomic Energy (58/14/03/2022-BRNS); Department of Biotechnology (BT/PR23498/BRB/10/1590/2017; BT/PR45508/MED/30/2414/2022); Council of Scientific and Industrial Research (CSIR) (37/1756/23/EMR-II); JC Bose fellowship (JCB/2021/000009), IISc-DBT partnership program (BT/PR27952/INF/22/212/2018) and infrastructure support provided by funding from DST-FIST and UGC are greatly acknowledged. DB was supported by a fellowship from the Department of Science and Technology and the Indian Institute of Science. HKD was supported by a fellowship from the Indian Institute of Science.

## Author contributions

**Debanjali Bhattacharya**: Conceptualization; Data curation; Formal analysis; Validation; Investigation; Methodology; Writing—original draft. **Harsh Kumar Dwivedi**: Data curation; Formal analysis; Investigation; Methodology; Writing—review and editing. **Ganesh Nagaraju**: Conceptualization; Resources; Supervision; Funding acquisition; Writing—original draft; Project administration.

Source data underlying figure panels in this paper may have individual authorship assigned. Where available, figure panel/source data authorship is listed in the following database record: biostudies:S-SCDT-10_1038-S44319-025-00497-3.

## Disclosure and competing interests statement

The authors declare no competing interests.

# Expanded View Figures

**Figure EV1. Loss of RNF20 leads to genomic instability in U2OS cells.**

(A) Representative images showing FANCD2 foci formation in control and RNF20-depleted mitotic U2OS cells following 22 h Aphidicolin treatment (0.4 µM) with RO3306 added in the last 6 h. Scale bar = 5 µm. Dotted white line indicates the nuclear content of the cells. (B) Quantified box and whiskers plot of FANCD2 foci per cell as indicated in (A). Mid-lines represent the medians and + represents mean values. Bounds of box represent 25th and 75th percentiles, whiskers represent 5–95 percentile. Total of ≥70 cells were analyzed for each condition from three experiments. Unpaired *t* test, **$P < 0.01$. $P$ (shRNF20 vs. shControl) = 0.0081. (C) Representative images showing pRPA32 S4/8 intensity in control and RNF20-depleted U2OS cells following 6 h recovery from HU (4 mM, 2 h) treatment. Scale bar = 5 µm. Dotted white line indicates the nuclear boundary of cells. (D) Quantitative bar plot of pRPA32 S4/8 intensity per cell in cells as indicated in (C). Data represents mean ± SEM from three independent experiments. Total of ≥300 cells were analyzed for each condition. Two-way ANOVA, *$P < 0.05$; **$P < 0.01$. $P$ (shControl UT vs. shRNF20 UT) = 0.0144, $P$ (shControl HU vs. shRNF20 HU) = 0.0014. (E) Representative images showing BrdU foci in the indicated cells after BrdU incorporation for 48 h followed by HU treatment (4 mM, 2 h). Scale bar = 5 µm. Dotted white line indicates nuclear boundary of the cells. (F) Quantitative bar plot showing percent of cells with BrdU foci in cells as shown in (E). Data represents mean ± SD from three independent experiments. Total of ≥250 cells were analyzed for each condition. Two-way ANOVA, *$P < 0.05$. $P$ (shControl UT vs. shRNF20 UT) = 0.0350, $P$ (shControl HU vs. shRNF20 HU) = 0.0488 (G) Representative images for alkaline comet assay in the indicated U2OS cells, either left untreated or treated with 4 mM HU for 2 h. Scale bar = 5 µm. (H) Scatter plot showing quantitative analysis of comet tail moments in indicated cells. Total of ≥250 cells were analyzed for each condition. Black bars represent median values. Mann–Whitney test, ****$P < 0.0001$. $P$ (shControl HU vs. shRNF20 HU) < 0.0001. (I) Representative images of metaphase spreads in the indicated cells with or without 2 mM HU treatment for 4 h. Arrows indicate breaks in chromosomes. Scale bar = 10 µm. (J) Quantitative plot showing number of chromosomal aberrations per spread, including breaks, gaps and radials, in cells as shown in (I). A minimum of 50 spreads were calculated for each sample. Data represents mean ± SD from three independent experimental repeats. Mann–Whitney test, ****$P < 0.0001$. $P$ (shControl HU vs. shRNF20 HU) < 0.0001. Source data are available online for this figure.

▶

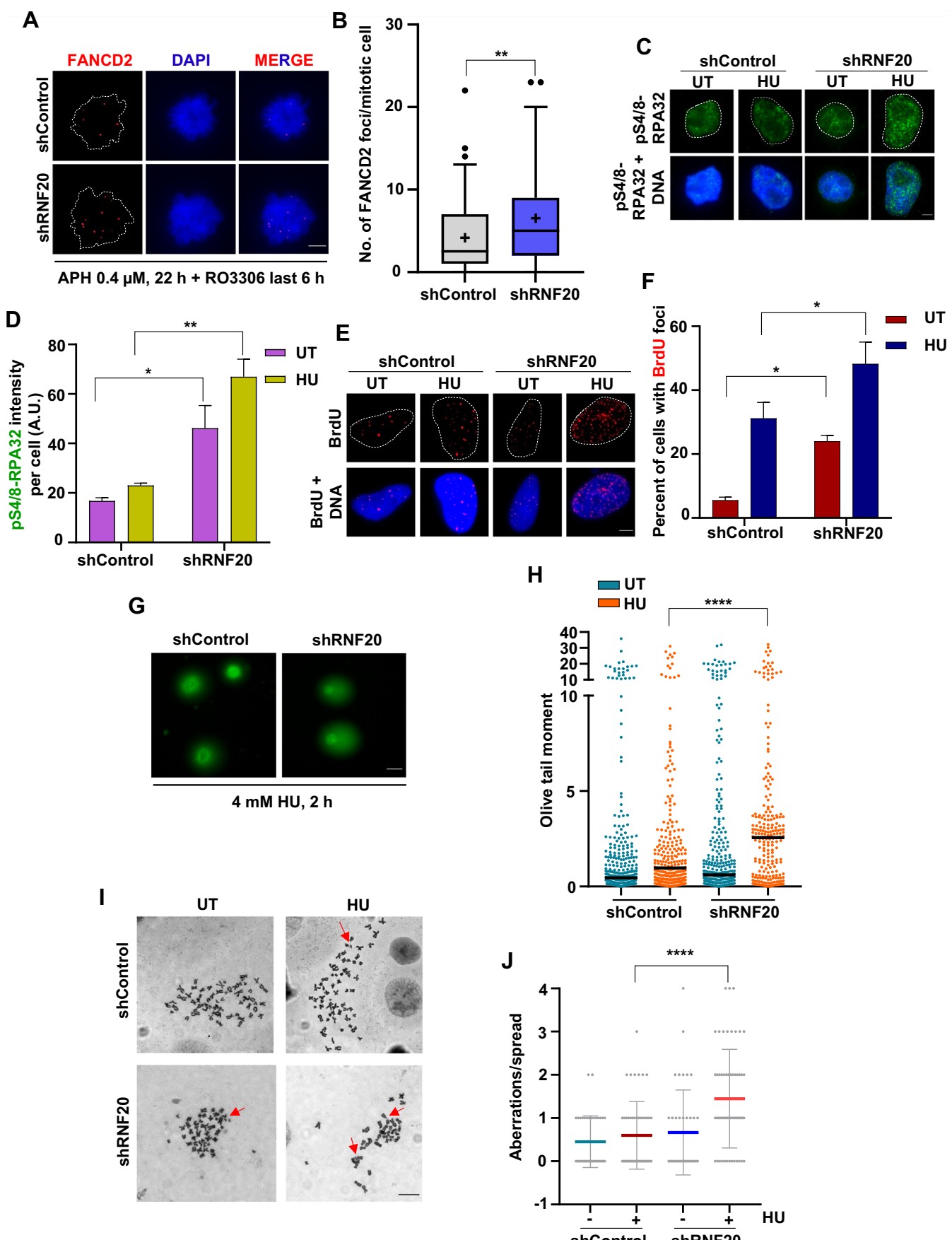

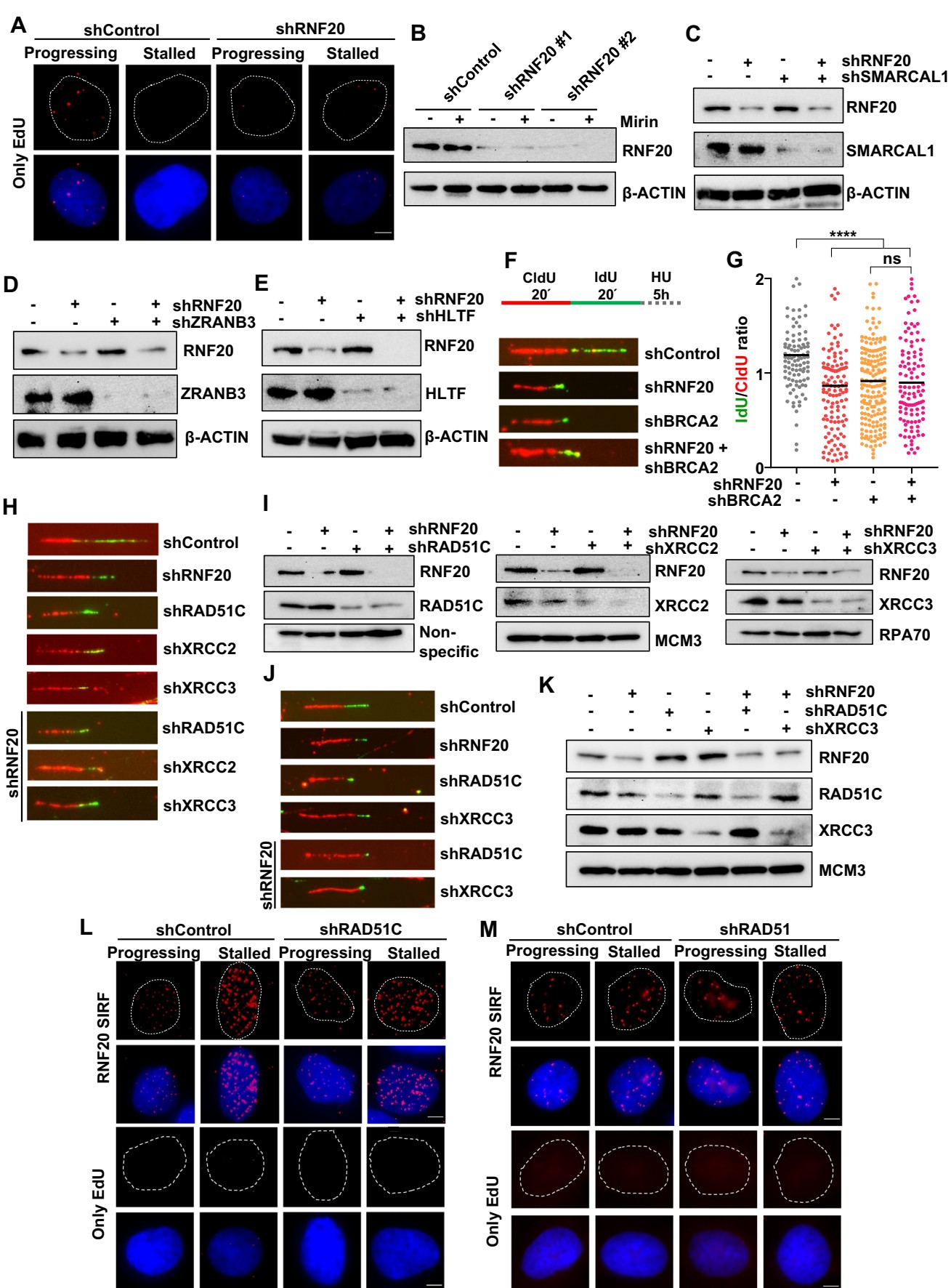

Figure EV2.   **RNF20 is epistatic to RAD51 paralogs in stalled fork protection and restart.**

(A) Representative images showing negative control of H2B K120ub SIRF (only EdU) for progressing and stalled replication sites in control and RNF20-depleted U2OS cells. Scale bar = 5 μm. Dotted white line indicates nuclear boundary of the cells. (B) Western blot showing depletion of RNF20 in U2OS cells transfected with shRNF20 #1 and #2 compared to shControl transfected cells with or without mirin treatment. (C) Western blot showing co-depletion of RNF20 and SMARCAL1 in U2OS cells. (D) Western blot showing co-depletion of RNF20 and ZRANB3 in U2OS cells. (E) Western blot showing co-depletion of RNF20 and HLTF in U2OS cells. (F) Scheme of DNA fiber labeling protocol (top) and representative DNA fibers showing fork stability in the indicated cells after 4 mM HU treatment for 5 h. (G) Quantitative scatter plot of IdU to CldU tract length ratio in cells as shown in (F). A total of ≥100 fibers were calculated for each sample from three experiments. Black bars represent median values. Mann–Whitney test, ****$P < 0.0001$. ns, non-significant. $P$ (shControl vs. shRNF20; shControl vs. shBRCA2; shControl vs. shRNF20 + shBRCA2) < 0.0001, $P$ (shRNF20 vs. shRNF20 + shBRCA2) = 0.4327. (H) Representative DNA fibers showing fork stability in the indicated cells treated with 4 mM HU for 5 h. (I) Western blot showing single and co-depletions of RNF20, RAD51C, XRCC2 and XRCC3 in U2OS cells transfected with the indicated shRNAs. (J) Representative DNA fibers showing replication fork restart in the indicated U2OS cells after release from 2 mM HU treatment for 2 h. (K) Western blot showing single and co-depletions of RNF20, RAD51C and XRCC3 in U2OS cells transfected with the indicated shRNAs. (L) Representative images of RNF20 SIRF signals at progressing and stalled replication forks in control and RAD51C-depleted U2OS cells. Scale bar = 5 μm. Dotted white line indicates the nuclear boundary of cells. (M) Representative images of RNF20 SIRF signals at progressing and stalled replication forks in control and RAD51-depleted U2OS cells. Scale bar = 5 μm. Dotted white line indicates the nuclear boundary of cells. Source data are available online for this figure.

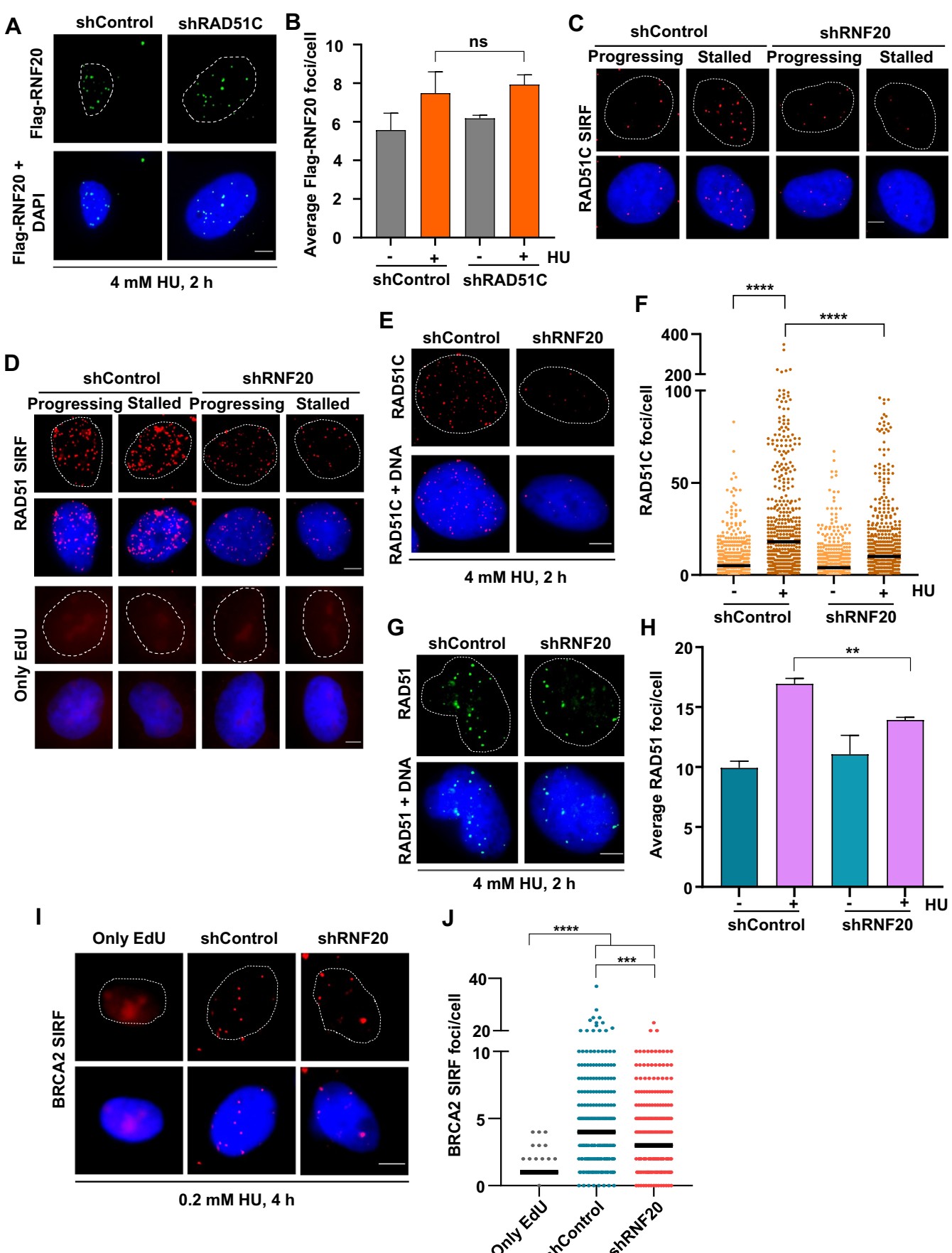

**Figure EV3. RNF20 facilitates the recruitment of RAD51/RAD51 paralogs to the stalled replication sites.**

(A) Representative images showing Flag-RNF20 foci in control and RAD51C depleted U2OS cells after 4 mM HU treatment for 2 h. Scale bar = 5 μm. Dotted white line indicates the nuclear boundary of cells. (B) Quantitative bar plot showing average Flag-RNF20 foci per cell in cells as shown in (A). Data represents mean ± SEM from three independent experiments. Total of ≥250 cells were analyzed for each condition. Unpaired $t$ test, ns, non-significant. $P$ (shControl HU vs. shRNF20 HU) = 0.7320. (C) Representative images of RAD51C SIRF signals in the indicated shRNA transfected U2OS cells in unperturbed (progressing) or HU treated (stalled) conditions. Scale bar = 5 μm. Dotted white line indicates the nuclear boundary of cells. (D) Representative images of RAD51 SIRF signals in the indicated shRNA transfected U2OS cells in unperturbed (progressing) or HU treated (stalled) conditions. Scale bar = 5 μm. Dotted white line indicates the nuclear boundary of cells. (E) Representative images showing RAD51C foci formation in control and RNF20-depleted U2OS cells during replication stress induced by 4 mM HU, 2 h treatment. Scale bar = 5 μm. Dotted white line indicates the nuclear boundary of cells. (F) Quantitative scatter plot showing RAD51C foci per cell for cells as shown in (E). Total of ≥300 cells were analyzed for each condition from three independent experiments. Black bars represent median. Mann–Whitney test, ****$P$ < 0.0001. $P$ (shControl UT vs. shControl HU; shControl HU vs. shRNF20 HU) < 0.0001. (G) Representative images showing RAD51 foci formation in control and RNF20-depleted cells subjected to HU treatment (4 mM, 2 h). Scale bar = 5 μm. Dotted white line indicates the nuclear boundary of cells. (H) Quantification of average RAD51 foci per cell as shown in (G). Data represents mean ± SEM from three independent experiments. Total of ≥250 cells were analyzed for each condition. Unpaired $t$ test, **$P$ < 0.01. $P$ (shControl HU vs. shRNF20 HU) = 0.0039. (I) Representative images of BRCA2 SIRF signals in the indicated shRNA transfected U2OS cells in HU treated (stalled) conditions. Scale bar = 5 μm. Dotted white line indicates the nuclear boundary of cells. (J) Quantitative scatter plot showing BRCA2 SIRF foci per cell for cells as shown in (I). Total of ≥300 cells were analyzed for each condition from three independent experiments. ≥110 cells were counted for only EdU control. Black bars represent median. Mann–Whitney test, ***$P$ < 0.001; ****$P$ < 0.0001. $P$ (shControl vs. shRNF20) = 0.0001, $P$ (Only Edu vs. shControl; Only Edu vs. shRNF20) < 0.0001. Source data are available online for this figure.

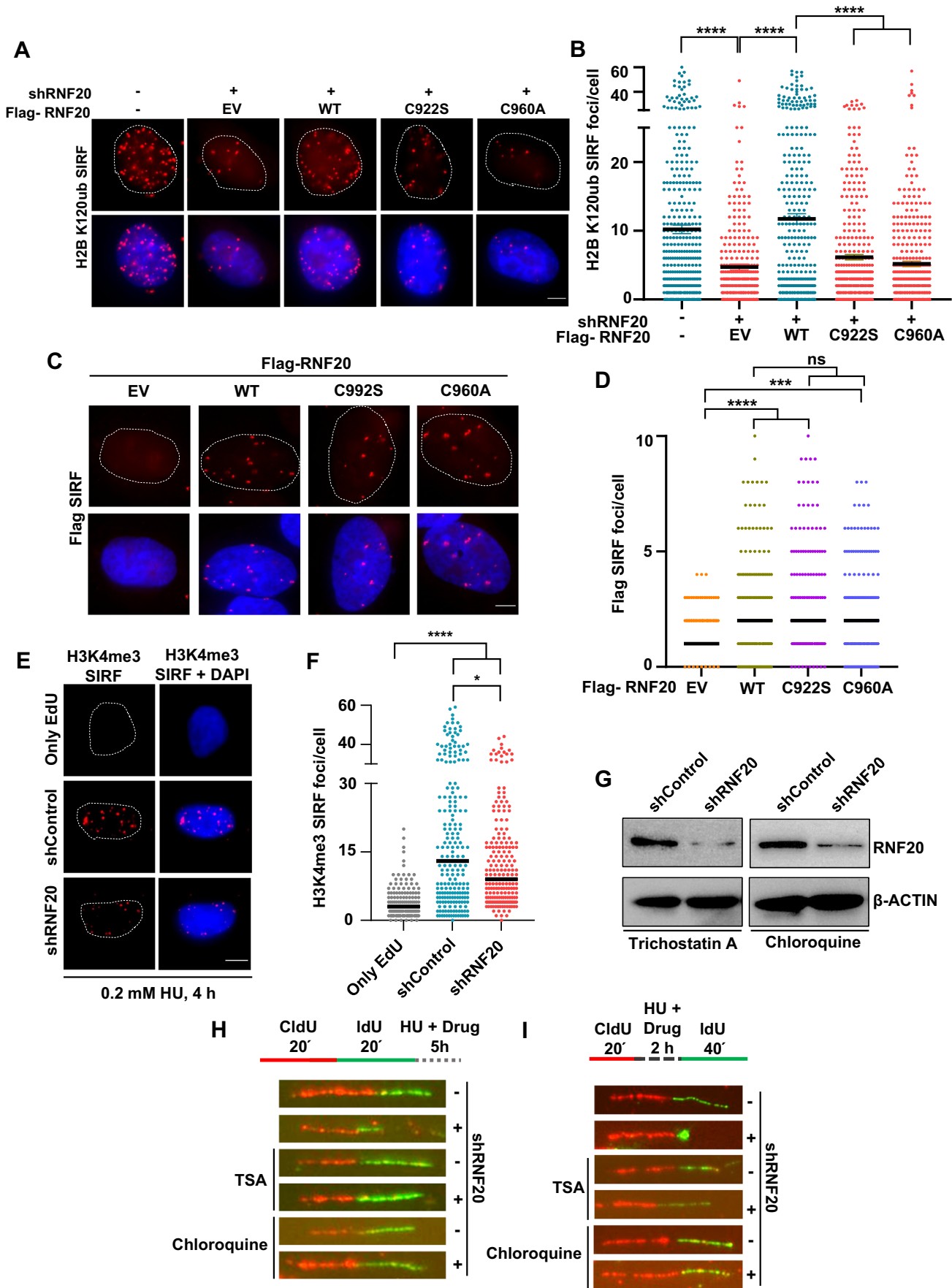

**Figure EV4. Catalytic activity of RNF20 is important for regulating chromatin dynamics at the stalled fork sites.**

(A) Representative images depicting H2B K120ub SIRF signals in U2OS cells expressing either WT or C922S/C960A RNF20 after endogenous RNF20 depletion. Scale bar = 5 μm. Dotted white line indicates the nuclear boundary of cells. (B) Scatter plot quantification of H2B K120ub SIRF signals in cells as shown in (A). Total of ≥250 cells were analyzed for each condition from three independent experiments. Data represents mean ± SEM. Mann–Whitney t test, ****P < 0.0001. P (shControl vs. shRNF20 + EV; shRNF20 + EV vs. shRNF20 + WT; shRNF20 + WT vs. shRNF20 + C922S; shRNF20 + WT vs. shRNF20 + C960A) < 0.0001. (C) Representative images of Flag-RNF20 SIRF signals in cells expressing WT or RING domain mutants of RNF20 at stalled fork sites. Scale bar = 5 μm. Dotted white line indicates the nuclear boundary of cells. (D) Scatter plot of Flag-RNF20 SIRF signals in cells as shown in (C). Total of ≥150 cells were analyzed for each condition from three independent experiments. Black bars represent median. Mann–Whitney t test, ***P < 0.001; ****P < 0.0001; ns, non-significant. P (EV vs. WT; EV vs. C922S) < 0.0001, P (EV vs. C960A) = 0.0003, P (WT vs. C922S) = 0.2296, P (WT vs. C960A) = 0.3429. (E) Representative images depicting H3K4me3 SIRF signals at stalled forks (0.2 mM HU, 4 h) in control and RNF20-depleted cells. Scale bar = 5 μm. Dotted white line indicates the nuclear boundary of cells. (F) Scatter plot quantification of H3K4me3 SIRF signals in cells as shown in (E). Total of ≥200 cells were analyzed for each condition from three independent experiments. Black bars represent median. Mann–Whitney t test, *P < 0.05; ****P < 0.0001. P (Only EdU vs. shControl; Only EdU vs. shRNF20) < 0.0001, (shControl vs. shRNF20) = 0.0122. (G) Immunoblot showing depletion of RNF20 in U2OS cells treated with HU (4 mM) + TSA (0.2 μM) or chloroquine (20 μg/ml) along with RNF20 shRNA. (H) DNA fiber labelling protocol (top) and representative DNA fibers showing fork protection in control and RNF20-depleted cells under HU (4 mM) and TSA (0.2 μM)/chloroquine (20 μg/ml) treatment. (I) DNA fiber labeling protocol (top) and representative DNA fibers depicting fork restart occurring in control and RNF20-depleted cells under HU (2 mM) and TSA (0.2 μM) or chloroquine (20 μg/ml) treatment. Source data are available online for this figure.

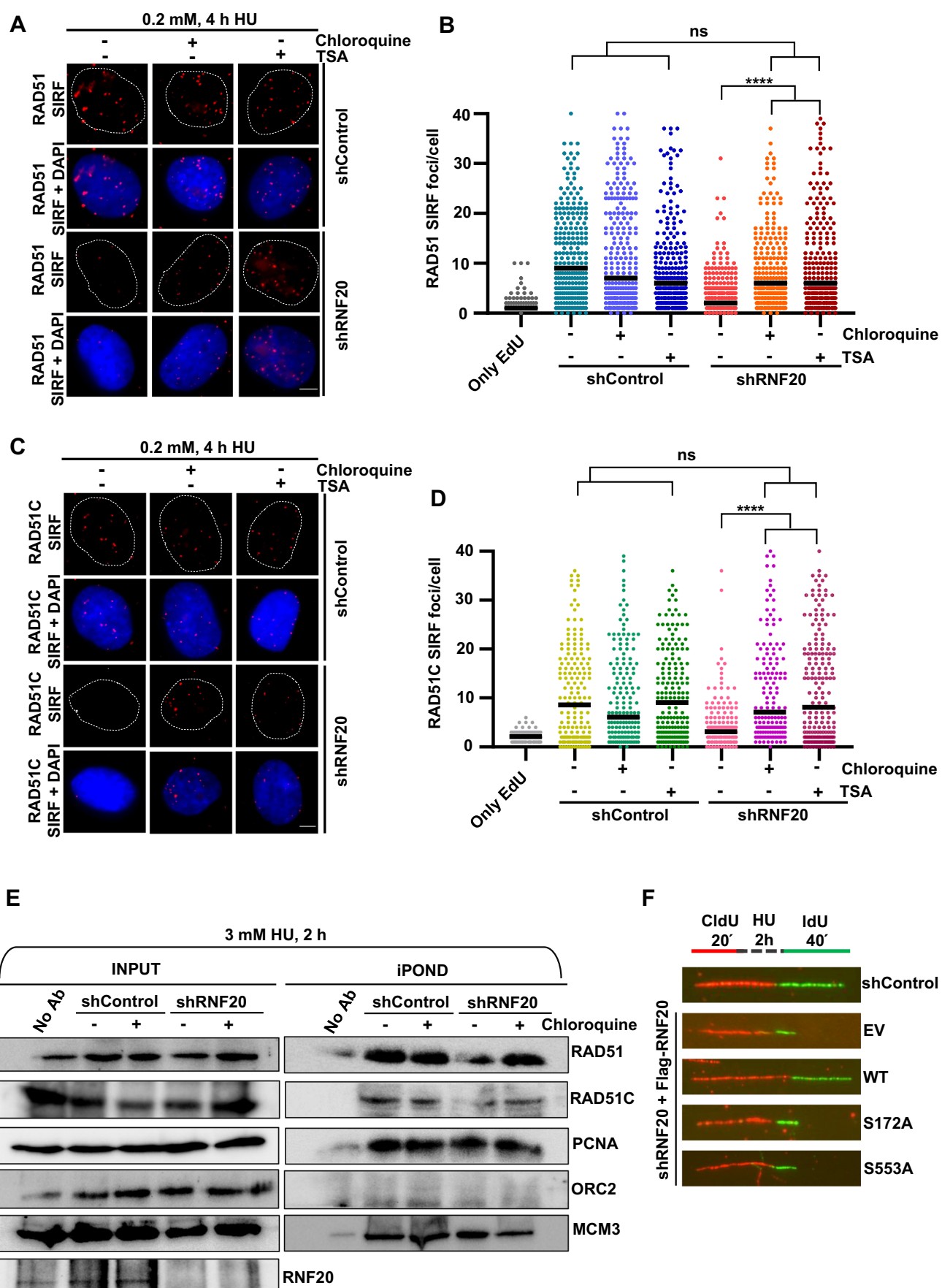

◀ **Figure EV5.** **Chromatin relaxation rescues the recruitment of RAD51/RAD51 paralogs to stalled forks in RNF20-deficient cells.**

(A) Representative images depicting RAD51 SIRF foci in control and RNF20-depleted U2OS cells treated with HU (0.2 mM) and TSA (0.2 µM) or chloroquine (20 µg/ml). Scale bar = 5 µm. Dotted white line indicates the nuclear boundary of cells. (B) Quantification of RAD51 SIRF foci in cells as shown in (A). Total of ≥250 cells were analyzed for each condition from three independent experiments. Black bars represent median. Mann–Whitney $t$ test, ****$P < 0.0001$. $P$ (shRNF20 vs. shRNF20 + chloroquine; shRNF20 vs. shRNF20 + TSA) < 0.0001. Kruskal–Wallis test with multiple comparisons, ns: non-significant. $P$ (shControl + chloroquine vs. shRNF20 + chloroquine) = 0.3593, $P$ (shControl + TSA vs. shRNF20 + TSA) > 0.9999. (C) Representative images depicting RAD51C SIRF foci in control and RNF20-depleted U2OS cells treated with HU (0.2 mM) and TSA (0.2 µM) or chloroquine (20 µg/ml). Scale bar = 5 µm. Dotted white line indicates the nuclear boundary of cells. (D) Quantification of RAD51C SIRF foci in cells as shown in (C). Total of ≥150 cells were analyzed for each condition from three independent experiments. Data represents median. Mann–Whitney $t$ test, ****$P < 0.0001$. $P$ (shRNF20 vs. shRNF20 + chloroquine; shRNF20 vs. shRNF20 + TSA) < 0.0001. Kruskal–Wallis test with multiple comparisons, ns: non-significant, $P$ (shControl + chloroquine vs. shRNF20 + chloroquine) >0.9999, $P$ (shControl + TSA vs. shRNF20 + TSA) > 0.9999. (E) Western blots showing localization of indicated proteins at stalled replication forks in control and RNF20-depleted cells by IdU-iPOND. Control and RNF20-depleted cells were pulsed with IdU followed by treatment with 3 mM HU for 2 h. IdU IP was performed and elutes were resolved by SDS-PAGE and blotted with indicated antibodies. (F) DNA fiber labeling protocol (top). Representative DNA fibers showing fork stalling and restart in the indicated U2OS cells after release from 2 mM HU, 2 h treatment (bottom). Source data are available online for this figure.

