## [Peer Review File · EMBO Reports]

RNF20-mediated H2B monoubiquitination protects stalled forks and promotes fork restart

Ganesh Nagaraju, Debanjali Bhattacharya, and Harsh Dwivedi

Corresponding author(s): Ganesh Nagaraju (nganesh@iisc.ac.in)

Review Timeline:

Submission Date:	25th Nov 24
Editorial Decision:	7th Jan 25
Revision Received:	22nd Apr 25
Editorial Decision:	9th May 25
Revision Received:	21st May 25
Accepted:	28th May 25

Editor: *Esther Schnapp*

Transaction Report:

Dear Prof. Nagaraju,

Thank you for the submission of your manuscript to EMBO reports. I could only secure 2 referees for your ms so far, but given that both are in fair agreement that you should be given a chance to revise your ms, I am making a decision now based on the 2 reports in order to save some time.

As you will see, the referees acknowledge that the findings are potentially interesting. However, they also have several suggestions for how the study could be improved and I think all suggestions are good and should be addressed. Please let me know in case you disagree and we can discuss the exact revision requirements further, also in a video chat, if you like.

I would thus like to invite you to revise your manuscript with the understanding that the referee concerns must be fully addressed and their suggestions taken on board. Please address all referee concerns in a complete point-by-point response. Acceptance of the manuscript will depend on a positive outcome of a second round of review. It is EMBO reports policy to allow a single round of major revision only and acceptance or rejection of the manuscript will therefore depend on the completeness of your responses included in the next, final version of the manuscript.

We realize that it is difficult to revise to a specific deadline. In the interest of protecting the conceptual advance provided by the work, we recommend a revision within 3 months (9th Apr 2025). Please discuss the revision progress ahead of this time with the editor if you require more time to complete the revisions.

- 1) A data availability section providing access to data deposited in public databases is missing. If you have not deposited any data, please add a sentence to the data availability section that explains that.
- 2) Your manuscript contains statistics and error bars based on $n=2$. Please use scatter blots in these cases. No statistics should be calculated if $n=2$.

3) We replaced Supplementary Information with Expanded View (EV) Figures and Tables that are collapsible/expandable online. A maximum of 5 EV Figures can be typeset. EV Figures should be cited as 'Figure EV1, Figure EV2' etc... in the text and their respective legends should be included in the main text after the legends of regular figures.

5) a complete author checklist, which you can download from our author guidelines <https://www.embopress.org/page/journal/14693178/authorguide>. Please insert information in the checklist that is also reflected in the manuscript. The completed author checklist will also be part of the RPF.

6) Please note that all corresponding authors are required to supply an ORCID ID for their name upon submission of a revised manuscript (<https://orcid.org/>). Please find instructions on how to link your ORCID ID to your account in our manuscript tracking system in our Author guidelines <https://www.embopress.org/page/journal/14693178/authorguide#authorshipguidelines>

12) All Materials and Methods need to be described in the main text using our 'Structured Methods' format, which is required for all research articles. According to this format, the Methods section includes a Reagents and Tools Table (listing key reagents, experimental models, software and relevant equipment and including their sources and relevant identifiers) followed by a Methods and Protocols section describing the methods using a step-by-step protocol format. The aim is to facilitate adoption of the methodologies across labs. More information on how to adhere to this format as well as a downloadable template (.docx) for the Reagents and Tools Table can be found in our author guidelines:
<https://www.embopress.org/page/journal/14693178/authorguide#structuredmethods>.

An example of a Method paper with Structured Methods can be found here: <https://www.embopress.org/doi/full/10.1038/s44320-024-00037-6#sec-4>

I look forward to seeing a revised form of your manuscript when it is ready.

Yours sincerely,

Referee #1:

The manuscript by Bhattacharya et al. describes a function for the RNF20 protein in the processing and stabilization of stalled DNA replication forks. Roles were identified in fork protection and re-start that depend on the RING domain of RNF20 and its phosphorylation by ATM/ATR. Overall, the study is comprehensive, and the data are convincing. I have some suggestions for improvements to the study.

Specific Remarks:

- 1) In figure 1, the authors use aphidicolin for some experiments in which replication is perturbed and then use hydroxyurea for others. Why was this? No justification was presented. It would be better to do all experiments with a specific agent. Also, the use of 53BP1 foci (bodies) in G1 phase cells is not necessarily the best - and certainly not the only - way to define the presence of under-replicated genomic regions. If the authors wish to claim that loci remain under-replicated, they should use other established markers like FANCD2 foci and/or mitotic DNA synthesis.
- 2) I could not understand why the authors used the seemingly complex SIRF assay to analyze the localization of factors to replication sites. This method incorporates PLA, which is not the most reliable assay system and is prone to false positive results. It would be better to use conventional IF and/or iPOND analysis, which is the gold standard for analyzing protein recruitment to forks.
- 3) The standard of the writing was good overall, but there were several places where 'the' was omitted in a sentence - such as line 2 of paragraph 2 in page 15 - 'we employed DNA fiber assay' and 6 lines up from the bottom on p16 - 'performing fork restart assay'. Also, on page 17, 6 lines up, 'we parallelly' should be replaced with 'In parallel, we...'
- 4) A lot is made of the effects of RNF20 on chromatin 'relaxation'. This is logical given its role in modifying histones. However, the analysis of drug-induced chromatin relaxation utilized very non-specific drugs like chloroquine and trichostatin A. Ideally, alternative/orthogonal method(s) should be employed to try to provide more convincing evidence that it is really chromatin relaxation that is driving the effects seen. On a related note, I suggest that the authors tone down statements like that in the last line of the Results - '... which in turn drives chromatin relaxation for replication stress responses.'
- 5) The Discussion section doesn't need to be as long as it is - particularly given that much of the text really just repeats the results found. It would be better if it focused on discussing the results in the context of what is already known in the literature and in the context of a model for the action of RNF20.
- 6) Given that the study has many similarities with work done previously by Liu et al (2021) on the yeast homolog of RNF20, Bre1, it would be helpful for the readers if the authors were to compare and contrast their findings in human cells with the previous work performed in yeast.
- 7) The Liu et al. paper mentioned in point #6 has an incomplete reference in the reference list - and there are others like this that I could see - such as Mishra et al., and Sethi et al. The references need to be checked carefully.

Referee #2:

In this manuscript, the authors investigate the role of RNF20 in protecting stalled replication forks from degradation and promoting fork restart. They demonstrate that RNF20 localizes to replication sites and promotes H2B monoubiquitination (H2Bub), leading to chromatin decompaction. This facilitates the recruitment of RAD51 and RAD51C to stalled forks, thereby enabling RAD51/RAD51C-mediated fork protection and restart. The proposed model underscores the critical role of RNF20 in regulating chromatin dynamics to safeguard replicating genomes. Overall, the topic of this manuscript is intriguing, and the study is well-executed.

Suggested Revisions:

1. There appears to be an over-reliance on proximity ligation assay (PLA) experiments to support key conclusions (see, for example, bioRxiv: <https://www.biorxiv.org/content/10.1101/411355v2.full>). To strengthen the findings, iPOND experiments are recommended to validate the localization of RNF20 at stalled forks and confirm its role in RAD51/RAD51C recruitment.

2. While the authors demonstrate that the MRE11-specific inhibitor Mirin rescues nascent DNA degradation in RNF20-deficient cells, it is unclear whether a similar rescue effect is observed with the DNA2 inhibitor C5. Since both Mirin and C5 have been reported to suppress nascent DNA degradation in BRCA-deficient cells, testing the effect of C5 on RNF20-deficient cells would provide additional mechanistic insights.
3. The authors propose that RNF20-mediated H2Bub promotes chromatin relaxation, which is critical for RAD51/RAD51C recruitment to stalled forks. To validate this model, it would be valuable to test whether treatment with chromatin-modifying agents, such as chloroquine and trichostatin A (TSA), rescues RAD51/RAD51C accumulation at stalled forks in RNF20-depleted cells. This should be assessed using both SIF and iPOND experiments to confirm the findings.
4. The manuscript does not address whether RNF20 is involved in the recruitment of BRCA proteins to stalled forks. Given that other factors, such as BOD1L, have been shown to facilitate RAD51 recruitment to counteract DNA2-mediated degradation, the authors should discuss how RNF20 might collaborate with BRCA proteins and other fork protection factors at stalled forks. Speculating on how these proteins coordinate their activities to prevent uncontrolled degradation by multiple nucleases would enrich the discussion section.

Response to referees

Referee #1:

The manuscript by Bhattacharya et al. describes a function for the RNF20 protein in the processing and stabilization of stalled DNA replication forks. Roles were identified in fork protection and re-start that depend on the RING domain of RNF20 and its phosphorylation by ATM/ATR. Overall, the study is comprehensive, and the data are convincing. I have some suggestions for improvements to the study.

Authors: We thank the reviewer for appreciating our work and providing suggestions to improve our study.

Specific Remarks:

1) In figure 1, the authors use aphidicolin for some experiments in which replication is perturbed and then use hydroxyurea for others. Why was this? No justification was presented. It would be better to do all experiments with a specific agent. Also, the use of 53BP1 foci (bodies) in G1 phase cells is not necessarily the best - and certainly not the only - way to define the presence of under-replicated genomic regions. If the authors wish to claim that loci remain under-replicated, they should use other established markers like FANCD2 foci and/or mitotic DNA synthesis.

Authors: 53BP1 nuclear bodies are mainly formed at Common Fragile Sites (CFS), which are specific genomic loci that intrinsically experience higher replication stress and are prone to frequent formation of breaks and gaps during replication. The non-random occurrence of breaks at specific genomic sites when induced with aphidicolin (APH) treatment forms the basis of defining most CFS sites in the genome (Harrigan et al., JCB. 2011). Hence, many studies have used APH to score for 53BP1 nuclear bodies which represents under-replicated genomic regions (Durkin and Glover, Annu. Rev. Genet. 2007; Harrigan et al., JCB. 2011; Lukas et al., Nat. Cell Biol. 2010; Spies et al., Nat. Cell Biol. 2019; Sarni et al., Nat. Comm. 2020). Moreover, other replication-stress-inducing drugs like hydroxyurea (HU) are less efficient at inducing CFS, probably due to the differences in their modes of action (Durkin and Glover, Annu. Rev. Genet. 2007; Harrigan et al., JCB. 2011). Similarly, cells also show accumulation of micronuclei upon CFS expression with APH treatment, where under-replicated DNA from CFS sites gets mis-segregated during mitosis to form micronuclear bodies (Benitez et al., Cell Rep. 2023; Xu et al., PLoS One, 2011; Wilhelm et al., Nat. Comm., 2019). Hence, we have performed our experiments scoring 53BP1 nuclear bodies and micronuclei formation using APH treatment.

As suggested by the reviewer, we have performed the experiments with a low dose of hydroxyurea (100 μ M HU for 16 h) and measured the 53BP1 nuclear bodies and micronucleation. Similar to APH conditions, our data with HU showed elevated 53BP1 nuclear bodies and micronucleation upon depletion of RNF20 in U2OS cells (Rebuttal Figure R1A and R1B). However, we have retained our results with APH treatment in the main figure (Figures 1B, 1C, 1D and 1E) as it seems more accurate to use APH for these experiments. Presently, we have added an explanation for using APH in these experiments in the first section of the results.

Rebuttal figure R1

As suggested, to show the presence of under-replicated DNA in cells lacking RN20, we scored for FANCD2 foci in mitotic cells, a bona-fide marker of under-replicated DNA in mitosis (Chan et al., Nat. Cell Biol. 2009). We found a significant increase in FANCD2 foci per cell in RNF20-depleted mitotic cells as compared to control cells, indicating the persistence of under-replicated genomic loci in the absence of RNF20 (Figures EV1A and EV1B).

2) I could not understand why the authors used the seemingly complex SIRF assay to analyze the localization of factors to replication sites. This method incorporates PLA, which is not the most reliable assay system and is prone to false positive results. It would be better to use conventional IF and/or iPOND analysis, which is the gold standard for analyzing protein recruitment to forks.

Authors: We understand the referee's concern about PLA and the possibility of false positive results associated with it. Although iPOND is a highly valuable technique that allows study of replication transactions in high-resolution, it is also very laborious, requires copious amounts of starting material (a minimum of 100 million cells per sample) and advanced technical skills to perform this technique. Additionally, using a large, heterogeneous pool of starting material makes this technique blind to changes occurring at a single-cell level (Sirbu et al. 2012).

On the other hand, SIRF assay (in situ protein interactions at nascent and stalled replication forks) allows for efficient analysis of protein interactions at nascent replication forks on a single-cell level. Unlike iPOND, it is quantifiable, requires very little starting cell material and can be performed in any standard molecular biology lab (Roy et al. 2019). In the recent

years, many labs have employed this technique to show localization of specific factors at replication forks (Longo et al., Nat. Comm. 2023; Leriche et al., Cell Rep. 2023; Han et al., Nat. Comm. 2022; Lyu et al., EMBO J. 2020). For its ease of use and single-cell, easily quantifiable resolution, we have used this technique to study the localization of different proteins at active and stalled replication fork sites. We have kept a **negative control** (using only anti-biotin antibody) in each experiment for each sample set to make sure that the SIRF signals are indeed coming from binding of a specific protein at replication sites and are NOT non-specific signals.

Additionally, we have also performed immunofluorescence experiments to validate our SIRF experiment results. While Flag-RNF20 foci per cell remained unaltered upon RAD51C depletion (Figures EV3A and EV3B), both RAD51C and RAD51 foci per cell were significantly diminished with RNF20-depletion (Figures EV3E, EV3F, EV3G and EV3H). These observations are consistent with our SIRF results.

However, we agree that iPOND is still an important technique for visualizing proteins at replication fork sites. Indeed, by iPOND analysis, Wessel et al., Cell Reports, 2019 showed that RNF20 is present at actively replicating sites. We have cited this reference also in our manuscript. As suggested, we have performed EdU-iPOND to validate our observations. In agreement with our SIRF results, we could detect RNF20 at actively replicating sites, and it was enriched at stalled forks. This new data is presented in Figures 2A and 2B. Consistent with our previous study (Somyajit, et al., NAR, 2015), RAD51 and RAD51C were also detected prominently at stalled forks. RPA70, PCNA, H2B and H3 were probed as controls in iPOND as studied earlier (Sirbu et al., Nat. Prot. 2012, Dungrawala et al., Mol. Cell, 2015, Genoio et al., Mol. Cell, 2021). Together, these observations validated our previous findings and confirmed an enrichment of RNF20 at stalled replication sites.

Kindly also refer to our response to Referee 2, points 1 and 3, where we have discussed our results of RAD51/RAD51C recruitment to stalled forks in RNF20-depleted cells by IdU-iPOND.

3) The standard of the writing was good overall, but there were several places where 'the' was omitted in a sentence - such as line 2 of paragraph 2 in page 15 - 'we employed DNA fiber assay' and 6 lines up from the bottom on p16 - 'performing fork restart assay'. Also, on page 17, 6 lines up, 'we parallely' should be replaced with 'In parallel, we...'

Authors: We have gone through the text carefully and made the necessary changes.

4) A lot is made of the effects of RNF20 on chromatin 'relaxation'. This is logical given its role in modifying histones. However, the analysis of drug-induced chromatin relaxation utilized very non-specific drugs like chloroquine and trichostatin A. Ideally, alternative/orthogonal method(s) should be employed to try to provide more convincing evidence that it is really chromatin relaxation that is driving the effects seen. On a related note, I suggest that the authors tone down statements like that in the last line of the Results - '... which in turn drives chromatin relaxation for replication stress responses.'

Authors: We thank the referee for this important suggestion. We understand that drugs like chloroquine or trichostatin A (TSA) induce genome-wide chromatin relaxation (Toth et al., J. Cell Sci. 2004; Joshi et al., Int. J. Mol. Sci. 2024). Indeed, previous studies have shown that defects in DSB repair by HR/NHEJ mechanisms in RNF20-deficient cells can be rescued by treating the cells with chloroquine and TSA (Moyal et al., Mol. Cell. 2011; Nakamura et al., Mol. Cell. 2011; Oliveira et al., J. Cell Sci. 2014; Hou et al., Nuc. Acids Res. 2020). Thus, we

took a similar approach to probe into RNF20's mechanism of action in stalled fork protection and restart. As suggested and to further validate that RNF20-mediated H2Bub indeed promotes chromatin decompaction, we examined H3K27ac and H3K4me3 modifications which are markers for chromatin relaxation at stalled fork sites. H3K27ac physically induces chromatin relaxation by neutralizing the charges on DNA through acetylation and is frequently associated with active transcription (Zambrano et al., Nat. Rev. Genet. 2022). Another histone mark co-existing with H3K27ac at active transcription sites is H3K4me3, which relaxes chromatin structure by recruiting chromatin remodeling complexes (Wysocka et al., Nature, 2006). Our results indicate that H3K27ac is indeed present at stalled forks in control cells but is impaired in RNF20-depleted cells (Figures 5G and 5H). Similarly, H3K4me3 SIRF foci showed a modest but significant decrease in RNF20-deficient cells compared to control cells (Figures S5E and S5F). These results corroborate our earlier observations and point towards a possible chromatin relaxation role of RNF20 in replication stress responses.

As suggested, we have modified "chromatin relaxation" to chromatin remodeling.

5) The Discussion section doesn't need to be as long as it is - particularly given that much of the text really just repeats the results found. It would be better if it focused on discussing the results in the context of what is already known in the literature and in the context of a model for the action of RNF20.

Authors: Thank you for the suggestion. As suggested, in the revised version, we have deleted redundancy with the results section and modified appropriately, including the literature and explaining the mechanism of RNF20 in replication stress responses presented in the model (Figure 7).

6) Given that the study has many similarities with work done previously by Liu et al (2021) on the yeast homolog of RNF20, Bre1, it would be helpful for the readers if the authors were to compare and contrast their findings in human cells with the previous work performed in yeast.

Authors: We thank the referee for this suggestion. We have now included a section in the discussion comparing our studies to the previous work performed mainly in yeast by Liu et al (2021).

7) The Liu et al. paper mentioned in point #6 has an incomplete reference in the reference list - and there are others like this that I could see - such as Mishra et al., and Sethi et al. The references need to be checked carefully.

Authors: We have verified the reference list and corrected the incomplete references.

Referee #2:

In this manuscript, the authors investigate the role of RNF20 in protecting stalled replication forks from degradation and promoting fork restart. They demonstrate that RNF20 localizes to replication sites and promotes H2B monoubiquitination (H2Bub), leading to chromatin decompaction. This facilitates the recruitment of RAD51 and RAD51C to stalled forks, thereby enabling RAD51/RAD51C-mediated fork protection and restart. The proposed model underscores the critical role of RNF20 in regulating chromatin dynamics to safeguard replicating genomes. Overall, the topic of this manuscript is intriguing, and the study is well-executed.

Authors: We thank the referee for positive comments about our study and appreciate the suggestions to improve our study.

Suggested Revisions:

1. There appears to be an over-reliance on proximity ligation assay (PLA) experiments to support key conclusions (see, for example, bioRxiv: <https://www.biorxiv.org/content/10.1101/411355v2.full>). To strengthen the findings, iPOND experiments are recommended to validate the localization of RNF20 at stalled forks and confirm its role in RAD51/RAD51C recruitment.

Authors: Reviewer 1 also had a suggestion to validate our SIRF assay results by iPOND. We request this reviewer to read our responses to Referee 1, comment 2. However, as suggested, and to strengthen our observations, we have now performed EdU-iPOND assay to analyze the recruitment of RNF20 and other associated factors like RAD51 and RAD51C to active, stalled and mature (chase) replication forks (Sirbu et al., 2012). Indeed, we find that RNF20 is associated with active forks and is enriched at stalled replication sites. These new data is presented in Figures 2A and 2B. These observations correlate with our SIRF data in Figures 2C and 2D. RAD51 and RAD51C were also enriched at the stalled fork sites, indicating they serve important roles in taking care of replication stress problems. These results further support our previous study with iPOND analysis, where we showed that RAD51 and RAD51 paralogs localizes to stalled fork sites (Somyajit et al., NAR, 2015).

Although we were able to optimize and perform EdU-iPOND with wild-type HeLa cells to test the localization of different factors to replicating sites, there were certain technical difficulties in performing this assay in RNF20-depleted condition. Each sample in iPOND requires a minimum of 100 million cells as starting material. We use the electroporation method for shRNA transfection to transiently deplete any protein of interest in cells. Electroporation leads to extensive cell death with only 30-35% of cell survival after transfection. Moreover, surviving fraction further declines upon RNF20 depletion. To obtain 100 million cells in RNF20-depleted condition, it was required to transfect > 300 million cells for each condition, which was not feasible in our situation. We therefore performed an IdU-IP based iPOND that has been described previously (Petermann et al., Mol. Cell. 2010; Somyajit et al., NAR. 2015; Dixit et al., Cell Reports, 2024) to isolate proteins from nascent DNA at replication forks in RNF20-depleted cells. We labelled nascent DNA with a 30 min IdU (5-iodo-2-deoxyuridine) pulse, treated cells with HU to induce fork stalling, crosslinked proteins to DNA and separated the chromatin fraction from the soluble fraction. IdU labelled strands were later pulled down from the chromatin fraction using anti-IdU antibody and proteins were analysed by western blotting. Data presented in Figure EV5E shows that RAD51 and

RAD51C recruitment to stalled replication forks is defective in RNF20-depleted cells. This is in line with our data obtained by SIF analyses (Figures 4E and 4F).

2. While the authors demonstrate that the MRE11-specific inhibitor Mirin rescues nascent DNA degradation in RNF20-deficient cells, it is unclear whether a similar rescue effect is observed with the DNA2 inhibitor C5. Since both Mirin and C5 have been reported to suppress nascent DNA degradation in BRCA-deficient cells, testing the effect of C5 on RNF20-deficient cells would provide additional mechanistic insights.

Authors: We thank the referee for this suggestion. To test if DNA2 inhibitor C5 can rescue fork degradation in RNF20-depleted cells, we analyzed the fork protection by DNA fiber assay in control and RNF20-depleted cells treated with HU+C5. Our data shows that treatment with C5 inhibitor rescues fork protection defect in RNF20-deficient cells (Figures 3E and 3F), similar to mirin treatment. These data indicate that RNF20 protects stalled replication forks from both MRE11 and DNA2-mediated fork degradation.

3. The authors propose that RNF20-mediated H2Bub promotes chromatin relaxation, which is critical for RAD51/RAD51C recruitment to stalled forks. To validate this model, it would be valuable to test whether treatment with chromatin-modifying agents, such as chloroquine and trichostatin A (TSA), rescues RAD51/RAD51C accumulation at stalled forks in RNF20-depleted cells. This should be assessed using both SIF and iPOND experiments to confirm the findings.

Authors: To investigate whether treatment with chromatin-modifying agents can rescue RAD51/RAD51C accumulation at stalled fork sites in RNF20-depleted cells, we performed SIF studies. Data presented in Figure EV5A, EV5B, EV5C and EV5D show that RNF20-depleted cells treated with chloroquine and TSA indeed rescue localization of RAD51/RAD51C to the stalled fork sites. To validate our observations, we performed an IdU-IP based iPOND that has been described previously (Petermann et al., Mol. Cell. 2010; Somyajit et al., NAR. 2015). Data presented in Figure EV5E shows that RAD51 and RAD51C recruitment to stalled replication forks is defective in RNF20-depleted cells and, treatment with chloroquine rescued the localization defect of RAD51 and RAD51C at stalled fork sites in RNF20-deficient cells. These observations validate our model, indicating that RNF20 regulates chromatin dynamics to facilitate the loading of fork protection factors.

4. The manuscript does not address whether RNF20 is involved in the recruitment of BRCA proteins to stalled forks. Given that other factors, such as BOD1L, have been shown to facilitate RAD51 recruitment to counteract DNA2-mediated degradation, the authors should discuss how RNF20 might collaborate with BRCA proteins and other fork protection factors at stalled forks. Speculating on how these proteins coordinate their activities to prevent uncontrolled degradation by multiple nucleases would enrich the discussion section.

Authors: We thank the referee for this remark. To investigate the recruitment of BRCA2 to stalled forks in RNF20-deficiency, we performed the SIF assay. Our results showed a significant reduction in BRCA2 SIF foci at stalled forks in RNF20-depleted cells compared to control cells (Figures EV3I and EV3J), indicating that RNF20 functions upstream of the FA-BRCA pathway and plays a global role in replication stress responses by catalyzing H2Bub, and subsequent chromatin remodeling associated with this modification to facilitate recruitment of fork protection factors.

Additionally, we have now included a section in our discussion about how different proteins like RNF20 and BOD1L may coordinate their activities to protect stalled forks from the action of multiple nucleases.

Dear Prof. Nagaraju

Thank you for the submission of your revised manuscript. We have now received the enclosed reports from the referees and I am happy to say that both support its publication now. Only a few editorial requests will need to be addressed before we can proceed with the official acceptance:

- Please provide up to 5 keywords with your ms.
- The conflict of interest subheading needs to be renamed to "Disclosure and Competing Interests Statement"
- The author credits need to be removed from the ms file. All credits need to be entered during online ms submission.
- The REFERENCE format needs to be corrected: - et al should be used after 10 author names. Please use the EMBO reports reference style.
- Please also enter all funding information during online ms submission, currently only a single funder is entered.
- The 2 EV tables in the ms need to be removed and uploaded as separate EV Table files; the correct nomenclature should be Table EV1 and Table EV2; the callouts such as EV Table 1 and EV Table 2 need to be corrected in the ms file.
- The Reagents & Tools TABLE needs to be removed from the ms file and uploaded as a separate file.
- The manuscript sections should be in the following order: Title page - Abstract & Keywords - Introduction - Results - Discussion - Methods - Data Availability - Acknowledgments - Disclosure Statement & Competing Interests - References - Figure Legends - (Main Tables with legends if applicable) - Expanded View Figure Legends.
- Materials and methods should be just Methods
- EV Figure legends are missing the word "Figure" in each title of the legends. Please add.
- All Microscopy images are low resolution figure files and show pixelation issue caused by conversion. Please upload higher resolution images created from the captured 16-Bit tiff files with your final ms submission.

Figure Legends - Comments

- Please note that the exact p values are not provided in the legends of figures 1C, I, J, K; 2D, F; 3B, D, F, H; 4C-F; 5D, H, I; 6H, EV1 H, J; EV2 G, EV3 F, J; EV4 B, D, F; EV5 B, D. Please provide exact p-values, as reasonable.
- Please note that the box plots need to be defined in terms of minima, maxima, bounds of box and whiskers, and percentile in the legends of figure EV1 B
- Please note that scale bar and its definition are missing for figure EV1 I.
- Please note that the dotted borders are not defined in the legend of figures 1B, F; 2C, E, 6G, EV1 A, C, E; EV2 A, L, M; EV3 A, C, D, E, G, I; EV4 A, C, E; EV5 A, C. This needs to be rectified.
- Please note that the white arrows are not defined in the legend of figure 1D. This needs to be rectified.

I would like to suggest a few minor changes to the abstract. Please let me know whether you agree with this:

Chromatin modifications play an important role in transcription, DNA replication and repair. Nonetheless, whether histone modifications regulate replication stress responses remains obscure. Here, we show that RNF20 localizes to and promotes H2B monoubiquitination (H2Bub) at replicating sites. Knockdown of RNF20 leads to degradation of stalled forks by nucleolytic enzymes, which can be rescued by inhibition of MRE11/DNA2 and co-depletion of SMARCAL1/HLTF/ZRANB3 fork remodelers. RNF20 facilitates the loading of RAD51 and RAD51C at stalled fork sites and acts in the same pathway of RAD51/RAD51C-mediated fork protection and restart. Analyses with RING domain and phosphorylation-deficient mutants of RNF20 show that its catalytic activity and ATR-mediated phosphorylation are essential for its role in replication stress responses. Finally, treatment of RNF20-depleted cells with chromatin relaxing agents rescues fork protection and restart defects. Collectively, our study uncovers a role for RNF20-mediated H2Bub in regulating chromatin dynamics to safeguard replicating genomes.

EMBO press papers are accompanied online by A) a short (1-2 sentences) summary of the findings and their significance, B) 2-3 bullet points highlighting key results and C) a synopsis image that is exactly 550 pixels wide and 200-600 pixels high (the height is variable). The synopsis image should provide a sketch of the major findings, like a graphical abstract. Please note that text needs to be readable at the final size. Please send us this information along with the final manuscript.

Referee #1:

I believe that the authors have made a strong effort to address my previous criticisms of the manuscript. I would now recommend that the manuscript be accepted for publication.

Referee #2:

The authors have addressed my concerns. I recommend the manuscript for publication in EMBO Reports.

All editorial and formatting issues were resolved by the authors.

Prof. Ganesh Nagaraju
Indian Institute of Science
Biochemistry
CV Raman avenue
BANGALORE 560012
India

Dear Prof. Nagaraju,

I am very pleased to accept your manuscript for publication in the next available issue of EMBO reports. Thank you for your contribution to our journal.
